# Inpainting-Guided Policy Optimization for Diffusion Large Language Models

**Siyan Zhao**[1,2*]  **Mengchen Liu**[1]  **Jing Huang**[1]  **Miao Liu**[3]  **Chenyu Wang**[1]  **Bo Liu**[1]
**Yuandong Tian**[1]  **Guan Pang**[1]  **Sean Bell**[1]  **Aditya Grover**[2]  **Feiyu Chen**[1]

[1]Meta Superintelligence Lab    [2]UCLA    [3]Tsinghua University

## Abstract

Masked diffusion large language models (dLLMs) are emerging as promising alternatives to autoregressive LLMs, offering competitive performance while supporting unique generation capabilities such as *inpainting*. We explore how *inpainting* can inform RL algorithm design for dLLMs. Aligning LLMs with reinforcement learning faces an exploration challenge: sparse reward signals and sample waste when LLMs fail to discover correct solutions. While this inefficiency affects LLMs broadly, dLLMs offer a distinctive opportunity—their inpainting ability can guide exploration. We introduce IGPO (Inpainting Guided Policy Optimization), an RL framework that strategically injects partial ground-truth reasoning traces during online sampling. Unlike providing full solutions, inpainting steers exploration toward promising trajectory spaces while preserving self-generated reasoning, bridging supervised fine-tuning and reinforcement learning. We apply IGPO to group-based optimization methods such as GRPO, where exploration failures cause zero advantages and gradients. IGPO restores meaningful gradients while improving sample efficiency. We also propose supervised fine-tuning on synthetically rewritten concise traces that better align with dLLM generation patterns. With additional techniques including entropy-based filtering, our training recipe yields substantial gains across four mathematical benchmarks—GSM8K, Math500, AMC and Minerva—achieving new state-of-the-art results for full-attention masked dLLMs. Our code is available here: https://github.com/facebookresearch/igpo.

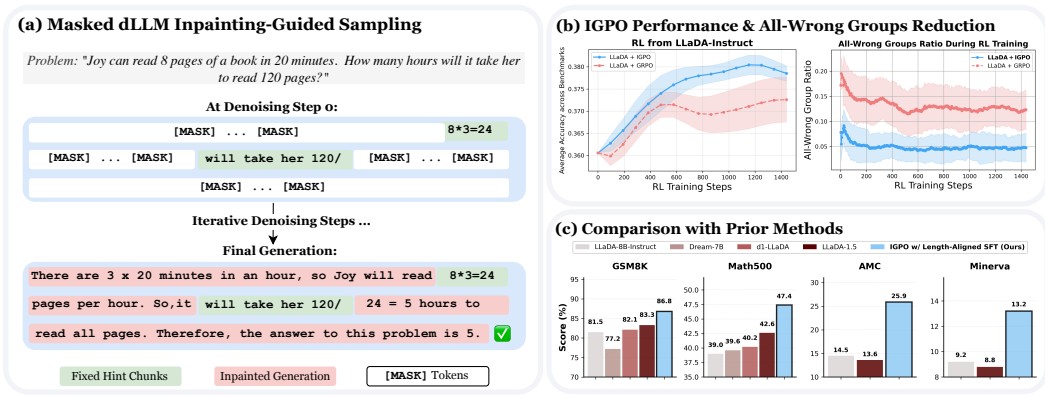

Figure 1: (a) Unlike autoregressive LLMs, diffusion LLMs can be conditioned on future reasoning hints during generation through *inpainting* via bidirectional attention, enabling guided exploration toward correct solutions. (b) Applying inpainting-guided exploration in policy optimization outperforms standard Group Relative Policy Optimization (GRPO) sampling and reduces all-wrong groups occurrences. (c) Our full training recipe combining *Length-Aligned* supervised fine-tuning on concise reasoning traces with IGPO achieves SoTA performance among full-attention masked dLLMs across four mathematical reasoning benchmarks.

---

*Work done at Meta

# 1 INTRODUCTION

Recent research has shown that masked diffusion large language models (dLLMs) (Austin et al., 2021; Lou et al., 2024; Shi et al., 2024) such as LLaDA (Nie et al., 2025) and Dream (Ye et al., 2025) can achieve performance competitive with autoregressive LLMs of similar size. Their capabilities and performance can be further enhanced via RL post-training (Zhao et al., 2025; Gong et al., 2025b) and ability to flexibly include multimodal data (Li et al., 2025; Yang et al., 2025; You et al., 2025). Unlike autoregressive LLMs, which decode in a left-to-right manner, dLLMs iteratively unmask tokens in parallel. This brings potential for faster inference as shown in closed models like Mercury (Inception Labs et al., 2025) and Gemini Diffusion (DeepMind, 2025), along with a flexible inductive bias for operations such as *inpainting*, the ability to fill in missing content within existing text.

In this work, we explore how *inpainting* can be leveraged to inform post-training algorithms for dLLMs. Recent work on post-training of dLLMs has adopted training approaches similar to autoregressive LLMs, applying Reinforcement Learning with Verifiable Reward (RLVR) methods (Zhao et al., 2025; Yang et al., 2025; Gong et al., 2025b). However, a fundamental exploration challenge persists: for challenging tasks, policies struggle to discover correct solutions and binary rewards provide minimal learning signal when most generated solutions are incorrect. This leads to substantial sample waste and poor training efficiency, exacerbating the computational costs of online RL.

The bidirectional generative structure of diffusion LLMs provides a unique mechanism to address this exploration challenge. Since dLLMs are trained through stochastic masking patterns, they possess inherent capability for accepting externally provided partial hints through *inpainting*. We leverage this ability to introduce IGPO (Inpainting Guided Policy Optimization), a novel RL framework that strategically guides exploration for dLLMs by injecting reasoning hints when answering difficult problems. Specifically, when the policy is unlikely to generate correct solutions, partial reasoning traces are injected into the generation region, and the dLLM is tasked with completing the remaining reasoning sequence and output final answer. The final answers are verified against ground truth, and only successful completions are used for downstream policy optimization.

We demonstrate IGPO's effectiveness in group-based policy optimization methods such as GRPO (Shao et al., 2024), which are particularly vulnerable to exploration failures: when a group's responses are all incorrect, group-normalized advantage collapses to zero and resulting in zero gradients. This occurs with alarming frequency in challenging domains. By reducing the prevalence of all-wrong groups, IGPO restores gradient signals and enables more effective RL. More broadly, IGPO can be viewed as a form of *guided exploration* that interpolates between supervised and RL paradigms. The injected tokens act as conditioning context that **steers the policy's action distribution toward high-reward regions**. Unlike pure SFT, which might suffer from distribution shift between data and policy rollouts (Zhang et al., 2025), IGPO maintains on-policy generation for the non-injected tokens. Finally, we augment IGPO with techniques that improve learning stability and performance, including entropy-based gradient filtering for injected tokens, and conduct comprehensive experiments across math benchmarks. We evaluate each component of our approach through ablation studies. Our work makes the following key novel contributions:

- We propose IGPO, the **first work to utilize the unique inpainting capabilities of diffusion LLMs** for reinforcement learning. By strategically injecting partial reasoning traces during exploration, IGPO alleviates the inefficiency of sparse verifiable rewards and mitigates the zero-advantage dilemma in group-based policy optimization methods, substantially reducing the proportion of all-wrong groups (by approximately 60% as shown in Fig 1 (b)) in our training.

- We propose a *Length-Aligned* SFT for full-attention based dLLMs using synthetically rewritten, concise reasoning traces. This design better aligns SFT data length with RL sampling and evaluation length, avoids the limitations of verbose traces, and provides stronger initialization for RL.

- Our full training recipe achieves substantial improvements on mathematical benchmarks, including **+5.3% on GSM8K, +8.4% on Math500, +11.4% on AMC, and +4.0% on Minerva** relative to the LLaDA-Instruct, achieving **SoTA performance among full-attention based dLLMs**.

- We conduct a comprehensive ablation study that disentangles the mechanisms of IGPO. We show that partial inpainting consistently outperforms full ground-truth inpainting by staying closer to the policy distribution in online RL, and propose an entropy-based gradient filtering mechanism that stabilizes training dynamics.

- We further show IGPO's robustness to imperfect reasoning traces and provide theoretical analysis of gradient recovery in the zero-advantage regime

## 2 PRELIMINARIES

### 2.1 MASKED DIFFUSION LARGE LANGUAGE MODELS

Masked diffusion LLMs (Austin et al., 2021; Sahoo et al., 2024; Shi et al., 2024; Ou et al., 2024; Lou et al., 2024) employ a forward diffusion (masking) process that progressively corrupts clean sequences $x_0$ by introducing mask tokens. This process is indexed by continuous time $t \in [0, 1]$. At any timestep $t \in (0, 1)$, the partially corrupted sequence $x_t$ is obtained by independently masking tokens so that each token remains unmasked with probability $\alpha_t$, where the schedule $\alpha_t$ is strictly decreasing in $t$. At $t = 1$, the sequence is fully masked. Training specifies the forward process via $\alpha_t$ and learns a bidirectional *unmasking predictor* $f_\theta$ to recover the original tokens from $x_t$. Each step samples $t \in [0, 1)$, applies the forward masking to obtain $x_t \sim q_{t|0}(x_t|x_0)$, and optimizes a masked-token objective derived from the negative evidence lower bound (NELBO), which upper-bounds the data negative log-likelihood (NLL). For masked dLLMs this NELBO reduces to a weighted NLL with weights determined by transforms of $\alpha_t$ (Sahoo et al., 2024, Eq. (10)). For example, LLaDA (Nie et al., 2025) uses a linear schedule $\alpha_t = 1 - t$, leading to:

$$-\mathbb{E}_{t\sim\mathcal{U}[0,1),\ x_0\sim p_{\text{data}},\ x_t\sim q_{t|0}(x_t|x_0)} \left[ \frac{1}{t} \sum_{k=1}^{|x_t|} \mathbb{1}[x_t^k = \texttt{mask}] \log f_\theta(x_0^k \mid x_t) \right], \quad (1)$$

where $|x_t|$ is the sequence length and $x^k$ the $k$-th token. The loss is computed only on tokens masked at time $t$. For prompt-conditional generation, prompt tokens are kept unmasked while continuation tokens are initialized as mask. The model then simulates a reverse process $p_\theta(x_s \mid x_t)$ over timesteps $t > s$, where $f_\theta$ provides denoising predictions for masked positions. Throughout the reverse trajectory, already unmasked tokens are preserved and carried forward unchanged.

### 2.2 POLICY OPTIMIZATION FOR MASKED DIFFUSION LARGE LANGUAGE MODELS

Policy-gradient post-training is widely used for LLM alignment (Ouyang et al., 2022; Bai et al., 2022; Li et al., 2023; Ahmadian et al., 2024). GRPO (Shao et al., 2024; Guo et al., 2025; Team et al., 2025) is a value-free variant of PPO (Schulman et al., 2017) that uses group-wise, sequence-level advantages for $G$ responses $\{o_i\}_{i=1}^G$ to a query $q$:

$$A_i = r(o_i) - \frac{1}{G} \sum_{j=1}^{G} r(o_j). \quad (2)$$

The GRPO objective integrates ratio clipping and reverse-KL regularization:

$$\mathcal{L}_{\text{GRPO}}(\theta) = \mathbb{E}_{\substack{q\sim\mathcal{D} \\ o_1,\dots,o_G\sim\pi_{\theta_{\text{old}}}(\cdot|q)}} \left[ \frac{1}{G} \sum_{i=1}^{G} \frac{1}{|o_i|} \sum_{k=1}^{|o_i|} \min\left(\rho_i^k A_i, \text{clip}\left(\rho_i^k, 1-\varepsilon, 1+\varepsilon\right) A_i\right) - \beta D_{\text{KL}}\left[\pi_\theta(\cdot|q)\|\pi_{\text{ref}}(\cdot|q)\right] \right], \quad (3)$$

where $\rho_i^k = \frac{\pi_\theta(o_i^k|q,o_i^{<k})}{\pi_{\theta_{\text{old}}}(o_i^k|q,o_i^{<k})}$ is the probability ratio, $r(\cdot)$ is a reward function, $\beta > 0$ is the KL regularization coefficient, $\varepsilon > 0$ is the clipping parameter, and $\pi_{\text{ref}}$ is the reference policy. In autoregressive models, the reverse-KL is tractable via the chain rule, $\log \pi_{\text{AR}}(o \mid q) = \sum_{k=1}^{|o|} \log \pi_{\text{AR}}(o^k \mid q, o^{<k})$, but masked diffusion LLMs do not admit a left-to-right factorization because $\pi_\theta$ arises from composing reverse denoising steps of the mask predictor. To make GRPO practical for masked diffusion policies, DiffuGRPO (Zhao et al., 2025) adopts a mean-field approximation that yields single-pass estimators for token-level ratios and the reverse-KL; we use these estimators throughout. We provide a detailed background discussion in Appendix B.

## 3 METHODS

### 3.1 IGPO: INPAINTING GUIDED POLICY OPTIMIZATION

**Zero-Advantage Dilemma.** In the GRPO framework, when sampling $G$ responses $\{o_1, o_2, \dots, o_G\}$ for a given prompt $q$, the advantage computation relies on reward variance across

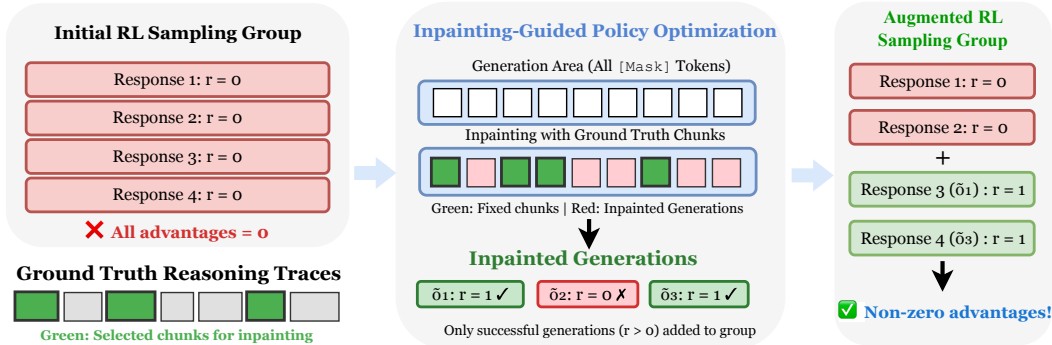

Figure 2: **Overview of IGPO:** When all sampled responses yield identical incorrect rewards (zero-advantage scenario), we perform hint-guided inpainting by generating additional responses using ground truth reasoning chunks as injected hints. Ground truth traces $y^*$ are segmented into variable-length chunks, and selected chunks are injected as fixed hints during generation while the model generates the remaining tokens. We then replace a fraction of the original incorrect responses with correct responses generated through inpainting, creating reward variance that enables non-zero advantages for effective policy gradient updates.

the group. However, when all responses receive identical rewards—either all correct or all incorrect—the advantages become zero: $A_i = r(o_i) - \frac{1}{G}\sum_{j=1}^{G} r(o_j) = 0$. This zero-advantage scenario makes the policy gradient component degenerate. Specifically, the clipped surrogate objective collapses to zero regardless of whether the update lies in the clipped or unclipped region, since both terms contain $A_i = 0$. The policy gradient for this prompt $q$ therefore becomes:

$$\frac{1}{G}\sum_{i=1}^{G}\frac{1}{|o_i|}\sum_{k=1}^{|o_i|} A_i\, \rho_i^k\, \nabla_\theta \log \pi_\theta(o_i^k \mid q) \;=\; 0 \qquad \text{since } A_i = 0\ \forall i.$$

As a result, no meaningful policy update can be extracted from the reward signal, wasting compute sampling these responses. **In this work, we specifically focus on mitigating the *all-wrong* case.**

**Masked dLLM Generation and Inpainting.** In full-attention masked dLLM generation, the model input at denoising step 0 is the concatenation $[q; z_{\text{mask}}]$, where $q$ represents the prompt and $z_{\text{mask}} = [\text{mask}, \text{mask}, \ldots, \text{mask}]$ denotes a fully masked completion sequence of predetermined length $L$. The generation process progressively unmasks these positions through iterative denoising until producing the final output.

*Hint injection* modifies this formulation by fixing selected positions of $z_{\text{mask}}$ to ground-truth tokens. During RL training, we assume access to ground-truth reasoning trace $y^* = [y_1^*, y_2^*, \ldots, y_{|y^*|}^*]$ for every question $q$. For injection, we create a binary mask $m \in \{0,1\}^L$ indicating which positions to inject as fixed hints, we construct the hint-injected initialization:

$$z^{\text{hint}}[i] = \begin{cases} y_i^* & \text{if } m[i] = 1 \text{ and } i \leq |y^*|, \\ \text{mask} & \text{otherwise.} \end{cases} \tag{4}$$

The masked dLLM then performs bidirectional denoising on $[q; z^{\text{hint}}]$ through the inpainting process, leveraging both the prompt and injected hint tokens to generate coherent responses. The injected hint tokens remain fixed throughout the iterative denoising steps.

**Constructing Hint Patterns for Inpainting.** To construct meaningful hint patterns for the inpainting process, we segment the ground truth reasoning trace $y^*$ into variable-length contiguous chunks $\mathcal{C} = \{c_1, c_2, \ldots, c_N\}$, where each chunk length $|c_j|$ is sampled from $\mathcal{U}[s_{\min}, s_{\max}]$. We explicitly exclude the final answer tokens from chunking to prevent reward hacking behaviors where the model ignores reasoning and collapses. For a given hint injection ratio $\eta \in [0, 1]$, we randomly select $\lfloor \eta \cdot N \rfloor$ chunks and set their corresponding positions in the binary mask $m$ to 1 for hint injection.

---

**Algorithm 1** IGPO: Inpainting-Guided Policy Optimization for Masked dLLMs

---

**Require:** Reference model $\pi_{\text{ref}}$, prompt distribution $\mathcal{D}$, ground-truth reasoning traces $\{y^*\}$, number of completions per prompt $G$, number of inner updates $\mu$, hint injection ratio range $[\eta_{\text{low}}, \eta_{\text{high}}]$, replacement fraction $\lambda$, entropy filter threshold $\tau$, chunk size range $[s_{\min}, s_{\max}]$

1: Initialize $\pi_\theta \leftarrow \pi_{\text{ref}}$
2: **while** not converged **do**
3:     $\pi_{\text{old}} \leftarrow \pi_\theta$; sample prompt $q \sim \mathcal{D}$ and responses $o_{1:G} \sim \pi_{\text{old}}(\cdot \mid q)$; compute rewards $r_{1:G}$
4:     **if** all $r_i = 0$ (zero-advantage case) **then**
5:         Segment ground-truth reasoning $y^*$ into chunks $\{c_1, \ldots, c_N\}$ with $|c_j| \sim \mathcal{U}[s_{\min}, s_{\max}]$
6:         **for** $i = 1, \ldots, G$ **do**
7:             Sample hint injection ratio $\eta \sim \mathcal{U}[\eta_{\text{low}}, \eta_{\text{high}}]$ and select $\lfloor \eta N \rfloor$ chunks from $\{c_1, \ldots, c_N\}$ randomly
8:             Inject selected chunk tokens as fixed hints at corresponding positions
9:             Generate $\tilde{o}_i$ via inpainting: denoise only masked positions, keep hint tokens fixed
10:         Evaluate rewards $r(\tilde{o}_i)$ and replace up to $\lfloor \lambda G \rfloor$ incorrect $o_i$ with correct $\tilde{o}_i$
11:     Compute advantages $A_i$ on the updated response set
12:     **for** $n = 1, \ldots, \mu$ **do**
13:         Estimate $\log \pi_\theta, \log \pi_{\text{old}}, \log \pi_{\text{ref}}$; apply top-$\tau$ entropy filter on hint positions
14:         Update $\pi_\theta$ via $\mathcal{L}_{\text{IGPO}}(\theta)$ (Eq. 5)
15: **return** $\pi_\theta$

---

**Elastic Inpainting-Triggered Sampling.** With the above inpainting setup, we design IGPO (as in Algorithm 1) to be **elastic**: hint injection is only triggered when all sampled responses in a group yield incorrect rewards (the zero-advantage case), and when activated, both the hint injection ratio $\eta$ and chunk sizes ($\mathcal{U}[s_{\min}, s_{\max}]$) are randomized to provide diverse training signals. Concretely, when detecting that all sampled responses $\{o_1, \ldots, o_G\}$ for query $q$ yield identical rewards $r(o_i) = 0$, we generate an additional set of responses $\{\tilde{o}_1, \ldots, \tilde{o}_G\}$ through the inpainting process. Each response $\tilde{o}_i$ is generated via inpainting with a distinct hint injection ratio $\eta_i \sim \mathcal{U}[\eta_{\text{low}}, \eta_{\text{high}}]$ to ensure diverse hint densities. Following inpainting generation, we evaluate the correctness of $\{\tilde{o}_i\}$ and only use the correct ones for replacement. Specifically, we replace $K = \min(|\{\tilde{o}_i : r(\tilde{o}_i) = 1\}|, \lfloor \lambda G \rfloor)$ of the original incorrect responses with correct responses generated through inpainting, where $\lambda \in (0, 1)$ controls the replacement fraction.

The only modification introduced by IGPO lies in the sampling step: when the all-wrong condition is detected, the original $G$ on-policy responses are partially replaced by $K$ correctness-verified inpainted samples. The IGPO objective therefore differs from the standard GRPO formulation only at the sampling level; all other components remain unchanged. In particular, the advantages $A_i$ are computed normally according to Eq. 2. The resulting objective for an all-wrong group is identical to GRPO except for this sampling change, highlighted in blue below.

$$\mathcal{L}_{\text{IGPO}}(\theta) = \mathbb{E}_{\substack{q \sim \mathcal{D} \\ \{o_1, \ldots, o_{G-K}, \tilde{o}_1, \ldots, \tilde{o}_K\} \sim \text{IGPO-Sample}(\pi_\theta, q, y^*)}} \left[ \left( \frac{1}{G} \sum_{i=1}^{G} \frac{1}{L_i} \sum_{k=1}^{L_i} \min\left(\rho_i^k A_i^k, \text{clip}\left(\rho_i^k, 1-\varepsilon, 1+\varepsilon\right) A_i^k\right) \right) - \beta D_{\text{KL}}\left[\pi_\theta(\cdot \mid q) \| \pi_{\text{ref}}(\cdot \mid q)\right] \right], \tag{5}$$

where IGPO-Sample$(\pi_\theta, q, y^*)$ denotes the augmented sampling procedure that applies inpainting-based augmentation when zero-advantage scenarios are detected, producing the augmented RL sampling group $\{o_1, \ldots, o_{G-K}, \tilde{o}_1, \ldots, \tilde{o}_K\}$ containing $(G - K)$ original responses and $K$ verified correct inpainted responses $\{\tilde{o}_i\}$ after replacement. $L_i$ denotes the length of the $i$-th response (whether $o_i$ or $\tilde{o}_i$). Crucially, only inpainted responses that pass correctness verification are included in the augmented group, satisfying $r(\tilde{o}_i) = 1$. We built IGPO with DiffuGRPO (Zhao et al., 2025)'s log probability estimation methods, where all completion tokens are masked during estimation and we remove the random masking applied to prompt tokens as done in DiffuGRPO. Since we use a small number of policy iterations (i.e. $\mu = 4$), this alleviates the need for random prompt masking to reduce overfitting. Inspired by Zheng et al. (2025), we compute sequence-level importance-ratio through mean-field approximation for stability purposes.

**Entropy-based Gradient Filtering for Hint Tokens.** When applying IGPO to zero-advantage scenarios, the responses generated through inpainting contain ground truth reasoning chunks that

originate from a different distribution than the current policy $\pi_\theta$. This creates an off-policy learning scenario where gradient updates from ground truth tokens can conflict with the model's current beliefs, particularly at positions where the model has high confidence (low entropy). To mitigate potential training instability from this distribution mismatch, we implement an entropy-based filtering approach that restricts learning to hint token positions where the model exhibits sufficient uncertainty, as inspired by Huang et al. (2025). Specifically, for each hint token position (i.e., positions with injected ground-truth tokens) we compute the entropy. We then apply gradient updates only to the top $\tau$ percentile of hint token positions with highest entropy values. This selective learning strategy serves two purposes: high-entropy positions represent genuine decision boundaries where the model is naturally uncertain and thus more receptive to external guidance, and they correspond to flatter probability distributions that yield more stable gradient updates when incorporating ground truth information. This approach controls the policy shift by focusing learning on positions where the model is already open to change, rather than forcing updates against strong existing beliefs at low-entropy positions.

### 3.2 LENGTH-ALIGNED SFT VIA CONCISE REASONING TRACE REWRITING

To further strengthen our training recipe, we seek better RL initialization via SFT but identified generation length mismatches across SFT, RL sampling, and evaluation phases. Full-attention masked dLLMs like LLaDA lack KV cache optimization (Wu et al., 2025) by defualt, requiring full-sequence attention at every denoising step, which dominates online RL training cost. As a result, we restrict RL rollouts to 256 tokens for faster convergence within a reduced exploration space, and evaluation setups in recent work (Zhao et al., 2025; Zhu et al., 2025; Nie et al., 2025) typically use 256–1024 tokens. In contrast, popular reasoning SFT corpora (e.g., OpenR1) contain verbose traces often exceeding 10k tokens, creating distribution mismatch across SFT, RL, and evaluation, and include repeated reflective behaviors unsuited for limited context. To resolve this, we systematically rewrite verbose traces into concise, structured forms that preserve logical flow while respecting dLLM computational limits. Using LLaMA-4-Maverick (Meta, 2025) with prompts detailed in Appendix I, we remove redundant reflections, condense multi-sentence elaborations into precise, mathematically rigorous statements, and retain essential reasoning. Examples of revision length distributions and before/after traces are in Appendix D and I. Our *Length-Aligned SFT* trains LLaDA solely on rewritten traces, improving RL initialization by avoiding implicit length compression and focusing learning on reasoning quality within fixed compute budgets. Empirical results show clear gains over training on verbose traces, and we further observe that masked dLLMs benefit from extended training (e.g., 100 epochs) relative to AR LLMs, consistent with recent works (Ni and the team, 2025; Prabhudesai et al., 2025).

## 4 EXPERIMENTS

To investigate how the inpainting capabilities of masked dLLMs can address the exploration challenges in RL and how *Length-Aligned* SFT improves performance, we conduct comprehensive experiments to answer the following main research questions:

(1) How effectively does our complete training approach (Length-aligned SFT with rewritten reasoning traces followed by reinforcement learning with IGPO) improve the mathematical reasoning performance of LLaDA and reduce all-wrong groups occurrences? (§4.3)
(2) How does partial hint injection in IGPO bridge on-policy generation with ground truth guidance, and how does this improve learning compared to full supervision? (§4.4)
(3) How do key design choices—including entropy filtering thresholds and reasoning trace rewriting—affect RL training dynamics and learning stability? (§4.4)

### 4.1 COMPLETE TRAINING RECIPE

Our complete learning framework consists of a two-stage pipeline: **Stage 1: Supervised Fine-Tuning with Rewritten Traces.** We begin with *Length-Aligned* SFT on the LLaDA-8B-Instruct model using the OpenR1-Math-220K dataset's default split (94k math problems), but with all reasoning traces rewritten (See Appendix D for length distribution before and after revision). This ensures consistency between training distribution and downstream RL/evaluation phases by aligning trace lengths. **Stage 2: Reinforcement Learning with IGPO.** Following *Length-aligned* SFT, we apply IGPO to further enhance reasoning capabilities through strategic inpainting-guided policy optimization. We utilize the reasoning traces from the MetaMathQA dataset for the elastic inpainting process, creating effective

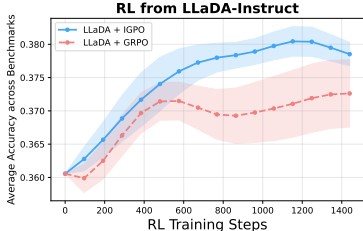 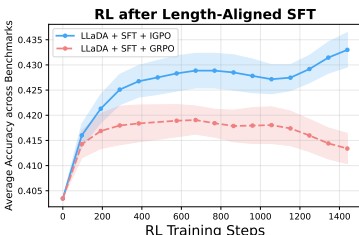

Figure 3: **RL training curves of IGPO versus normal GRPO sampling.** (a) Starting from LLaDA-8B-Instruct. (b) Starting from the *length-aligned* SFT checkpoint. IGPO exhibits superior and more stable training performance under both initialization checkpoints compared to standard GRPO sampling. Results are averaged over 3 random seeds across four mathematical reasoning benchmarks (GSM8K, MATH500, AMC and Minerva Math), with standard errors shown as shaded regions.

guidance signals that fit within our computational constraints. Detailed training hyperparameters are provided in Appendix F.

## 4.2 EXPERIMENTAL SETUP

We conduct experiments using LLaDA-8B-Instruct as the base model with a sampling temperature of 1.2 for RL online generation, where the temperature is selected based on exploration and exploitation analysis detailed in Appendix F.1. For reinforcement learning, we train on the MetaMathQA dataset (Yu et al., 2023), specifically using the "Answer Augmentation" split and combining questions from both GSM8K and MATH500. After deduplicating identical questions, we obtain 12,794 unique training examples. For supervised fine-tuning, we utilize the OpenR1-Math-220K dataset with rewritten reasoning traces as described in Section 3.2. We evaluate our approach on four mathematics benchmarks: GSM8K (Cobbe et al., 2021), MATH500 (Hendrycks et al., 2021), AMC (LI et al., 2024) and Minerva Math (Lewkowycz et al., 2022). Experiments are conducted on 8×8 80GB H100 GPUs. For UniGRPO (Yang et al., 2025) baseline, we reproduce based on their Algorithm 1. We provide detailed experiment hyperparameter setups in Appendix F and Appendix H.

## 4.3 MAIN RESULTS

Table 1: Performance across multiple mathematics tasks. GSM8K, MATH500 and Minerva are evaluated with pass@1 at temperature of 0.0, and AMC with avg@16 at temperature 0.1. Underlined scores indicate the best *within each initialization group*. Parenthesized deltas typeset via (+) denote absolute percentage-point improvements *relative to the LLaDA-8B-Instruct* baseline.

| Model | GSM8K (pass@1) | MATH500 (pass@1) | AMC (avg@16) | Minerva (pass@1) | Average |
|---|---|---|---|---|---|
| *Similar-sized autoregressive LLMs* | | | | | |
| LLaMA3-8B (AI@Meta, 2024) | 79.6 | 30.0 | – | – | – |
| Qwen2.5-7B (Team, 2024) | 85.4 | 49.8 | – | – | – |
| *Prior masked dLLM baselines* | | | | | |
| Dream-7B (Ye et al., 2025) | 77.2 | 39.6 | – | – | – |
| d1-LLaDA (Zhao et al., 2025) | 82.1 | 40.2 | – | – | – |
| wd1 (Tang et al., 2025) | 82.3 | 39.0 | – | – | – |
| LLaDA-1.5 (Zhu et al., 2025) | 83.3 | 42.6 | 13.6 | 8.8 | 37.1 |
| LLaDA-Instruct (Nie et al., 2025) | 81.5(+0) | 39.0(+0) | 14.5(+0) | 9.2(+0) | 36.0(+0) |
| *RL from LLaDA-Instruct* | | | | | |
| LLaDA-Instruct + UniGRPO (Yang et al., 2025) | 82.2(+0.7) | 39.2(+0.2) | 15.0(+0.5) | 11.0(+1.8) | 36.9(+0.9) |
| LLaDA-Instruct + DiffuGRPO (Zhao et al., 2025) | 82.4(+0.9) | 40.2(+1.2) | 15.5(+1.0) | 10.3(+1.1) | 37.1(+1.1) |
| LLaDA-Instruct + IGPO (ours) | 83.1(+1.6) | 42.8(+3.8) | 17.5(+3.0) | 12.1(+2.9) | 38.9(+2.9) |
| *Length-aligned SFT on LLaDA-Instruct and RL on the SFT checkpoint* | | | | | |
| LLaDA-Instruct + Length-aligned SFT (ours) | 83.6(+2.1) | 45.2(+6.2) | 22.3(+7.8) | 10.3(+1.1) | 40.4(+4.4) |
| LLaDA-Instruct + Length-aligned SFT + IGPO (ours) | **86.8**(+5.3) | **47.4**(+8.4) | **25.9**(+11.4) | **13.2**(+4.0) | **43.3**(+7.3) |

As shown in Table 1, our training recipe demonstrates consistent improvements across all mathematical reasoning benchmarks. With *Length-Aligned* SFT on rewritten traces, LLaDA achieves an average improvement of 4.4% compared to the base LLaDA-8B-Instruct model. When applying IGPO on top of the SFT model, we observe additional improvements, resulting in a total average improvement of 7.3%. The complete two-stage pipeline yields cumulative improvements of 5.3% on GSM8K, 8.4% on MATH500, 11.4% on AMC, and 4.0% on Minerva relative to the LLaDA-Instruct baseline. Notably, on the challenging AMC benchmark, our approach achieves 25.9% (avg@16). As shown in Figure 3, IGPO exhibits superior training dynamics compared to standard GRPO sampling when initializing from before or after SFT. IGPO effectively reduces the all-wrong group ratio by approximately 60%, as shown in Figure 1(b). Our final model (LLaDA + *Length-Aligned* SFT + IGPO) outperforms all baseline approaches including the recent LLaDA-1.5 model across all evaluated benchmarks. Notably, even without SFT, applying IGPO directly on LLaDA achieves better performance than the previous LLaDA-1.5 and other RL methods for full-attention dLLMs, establishing a new state-of-the-art recipe for mathematical reasoning in masked diffusion language models.

## 4.4 ANALYSIS AND ABLATION STUDIES

**Self-generated inpainted traces provide better learning signal than ground truth traces.** The results in Figure 4 show that partial hint injection achieves higher performance than full hint injection. When the hint injection ratio varies within the lower range, the model needs to generate self-rationalized inpainting traces (with an example shown in Appendix G), and only those that lead to correct solutions are added to the group for gradient updates. Through inpainting, the model attempts to coherently connect provided hint chunks with its own reasoning steps. The inpainted generation produces a learning signal that bridges the gap between the model's current capabilities and the target behavior. The self-generated portions reflect the model's current reasoning patterns and are more "on-policy" while incorporating structural guidance from ground truth chunks, resulting in more effective policy optimization compared to pure supervised learning, reducing the distributional mismatch. **This bridging of SFT and online RL through partial self-generation leads to more effective policy optimization.**

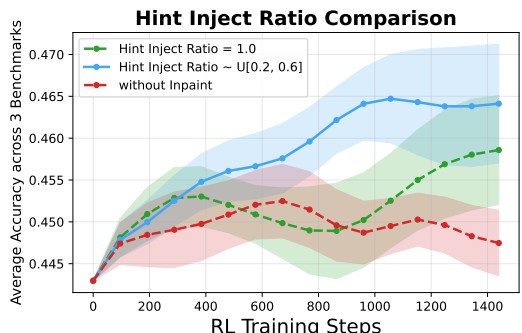

Figure 4: **Impact of hint injection ratio.** across 3 datasets (GSM8K, MATH500 and AMC) and 3 seeds with standard error shown as shaded areas. We compare partial hint injection ($\eta \sim \mathcal{U}[0.2, 0.6]$) versus full hint injection ($\eta = 1.0$). Partial hint injection consistently outperforms full hint injection, demonstrating the benefits of self-generated reasoning. Both hint-guided inpainting variants outperform the baseline without any hint injection.

**Entropy clipping prevents training instability from off-policy tokens.** As shown in Figure 5a, we observe that learning from only the top 20% highest-entropy hint token positions ($\tau = 0.2$) achieves the best performance and exhibits the most stable training dynamics. In contrast, learning from all hint token positions ($\tau = 1.0$) or a large fraction ($\tau = 0.8$) leads to more unstable training with performance fluctuations compared to lower values like 0.2. This empirical finding supports our motivation that restricting gradient updates to high-entropy positions prevents the destabilizing effects of large gradients on high-entropy positions. The validates the necessity of entropy-based filtering when incorporating ground truth traces from hint-guided inpainting into policy gradient training.

**Effect of reasoning trace rewriting for SFT and subsequent RL training.** The results in Figure 5b illustrate two key findings. First, SFT on rewritten reasoning traces produces substantially stronger checkpoints than SFT on the original traces. Our rewritten traces eliminate verbose reflection behaviors and compress reasoning into concise trajectories (up to 1024 tokens), which are better aligned with LLaDA's generation budget (256 tokens) and evaluation sequence length. This alignment improves SFT accuracy at step 0 relative to models trained on the longer 4096-token traces. Second, while RL training can partially compensate for weaker SFT checkpoints—the models trained on 4096-token traces recover accuracy rapidly in early RL steps—starting from stronger rewritten SFT

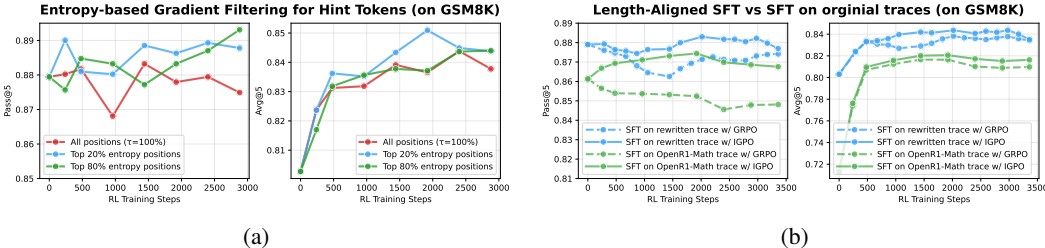

(a)                                                    (b)

Figure 5: (a) **Impact of entropy clipping threshold on hint tokens.** Performance comparison across different entropy clipping thresholds $\tau$ applied to hint token positions in IGPO, where $\tau = 0.2$ represents learning from only the top 20% highest-entropy hint token positions, while $\tau = 1.0$ indicates learning from all hint token positions without filtering. (b) **SFT and RL dynamics with rewritten vs. original traces.** We compare models fine-tuned on concise rewritten traces (max 1024 tokens) vs on original OpenR1-Math traces truncated at LLaDA's 4096 context limit. RL is then applied (GRPO or IGPO) to both models. Rewritten traces yield stronger SFT performance and superior RL outcomes. IGPO consistently outperforms GRPO with stable pass@5 while GRPO suffers from diversity collapse. Results are run on GSM8K with temperature 0.1 and length 256.

checkpoints leads to consistently higher final performance. Importantly, across both initialization settings, IGPO outperforms standard RL without inpainting. Additionally, IGPO preserves output diversity and stabilizes pass@5 performance throughout training, whereas standard GRPO exhibits a decline in pass@k metrics, indicative of reduced exploration and mode collapse.

**Elastic inpainting outperforms sequential SFT and GRPO**   We further validate the effectiveness of our elastic inpainting approach by comparing it against sequentially performing SFT on the RL dataset's reasoning traces followed by standard GRPO (see Appendix E for details). This ablation confirms that IGPO's elastic hint injection during zero-advantage scenarios is superior to uniformly applying SFT on concise reasoning traces across all prompts before applying GRPO. The uniform SFT approach can degrade initial performance due to distribution shift in reasoning patterns, whereas injecting partial hints allows dLLMs to inpaint longer, more "on-policy" reasoning traces.

**Robustness to noisy ground-truth reasoning traces.**   We further investigate how IGPO behaves when the injected ground-truth reasoning traces are imperfect. Following the setup described in Appendix J, we introduce synthetic corruption into reasoning traces by randomly perturbing tokens with probability $\rho$ (including numerical mistakes, operator swaps, and logical inconsistencies). As shown in Appendix J and Figure 9, IGPO maintains strong performance even under moderate noise levels and consistently outperforms GRPO across all corruption rates. Performance gradually degrades as $\rho$ increases, but the advantage of IGPO remains stable. This robustness arises from two design choices: only partial reasoning chunks are injected during inpainting and updates are gated by correctness verification and entropy filtering, which limits the impact of incorrect hint tokens. These results suggest that IGPO can effectively leverage imperfect reasoning traces without requiring perfectly clean supervision.

**Comparison to compute-matched GRPO.**   To determine whether IGPO's improvements arise from its hint-guided exploration rather than increased sampling, we compare against a compute-matched GRPO baseline that uses the same sampling budget. As detailed in Appendix L, the baseline (GRPO + resample) generates additional trajectories whenever a group produces only incorrect responses, matching the total number of samples used by IGPO while remaining fully on-policy. Despite using the same number of trajectories, GRPO + resample exhibits a higher all-wrong ratio and lower training rewards, while IGPO consistently achieves better evaluation performance (see Figures 10 and 11). These results demonstrate that IGPO's gains do not stem from increased compute or sampling volume, but from its targeted inpainting-based guidance, which provides directional exploration signals that pure resampling cannot replicate.

## 5 RELATED WORK

**Diffusion Language Models:** Recent advances in diffusion language models have progressed from continuous approaches mapping discrete text to continuous representations (Chen et al., 2022; Li et al., 2022; Gong et al., 2023) to scaled discrete diffusion models, with masked diffusion emerging as a prominent approach (Austin et al., 2021; Sahoo et al., 2024; Shi et al., 2024; Ou et al., 2024; Nie et al., 2024). Notable developments include DiffuLLaMA (Gong et al., 2025a) and Dream (Ye et al., 2025) adapted from pretrained autoregressive LLMs, and LLaDA (Nie et al., 2025) as a masked diffusion LLM trained from scratch achieving comparable performance to autoregressive models. Commercial models like Mercury (Inception Labs et al., 2025) and Gemini Diffusion (DeepMind, 2025) have demonstrated practical viability with significantly faster inference. **Reinforcement Learning for Diffusion Language Models:** Applying RL to diffusion LLMs faces unique challenges due to intractable likelihood estimation, which is required for policy optimization. Recent solutions include diffu-GRPO (Zhao et al., 2025) with mean-field approximation, MMaDA (Yang et al., 2025) and coupled-GRPO (Gong et al., 2025b) with improved masking strategies, LLaDA 1.5 (Zhu et al., 2025) addressing variance through preference optimization, wd1 (Tang et al., 2025) eliminating policy ratios via weighted likelihood objectives, and SDPO (Han et al., 2025) decomposing trajectory alignment into stepwise subproblems. More detailed related works are discussed in Section C.

## 6 CONCLUSION

We introduced IGPO, a reinforcement learning algorithm that leverages the inpainting capabilities of masked diffusion language models. By injecting ground-truth reasoning hints during denoising, IGPO steers the policy toward high-reward regions and alleviates the exploration bottleneck in RL. It resolves the zero-advantage dilemma by inducing reward variance that supports effective policy gradient updates when standard sampling yields uniform outcomes. To further strengthen RL initialization, we proposed *Length-Aligned* SFT, which reduces the length mismatch across SFT, RL, and evaluation stages. Combined with entropy-based gradient filtering, our approach achieves new state-of-the-art performance among full-attention masked dLLMs on multiple mathematical reasoning benchmarks. These results highlight a new paradigm for reinforcement learning in masked diffusion language models, showing how architectural properties can be systematically exploited to address critical optimization challenges.

## ETHICS STATEMENT

This work focuses on algorithmic contributions to reinforcement learning and supervised fine-tuning for diffusion language models, specifically targeting mathematical reasoning tasks. Our research does not involve human subjects, does not collect or release new datasets containing personal information, and addresses computational methods for improving model performance on well-established mathematical benchmarks. The mathematical reasoning domain we target does not raise concerns about harmful applications, bias amplification, or misuse potential. Our methodology improvements are designed to enhance model accuracy and training efficiency, which we believe contributes positively to the field of AI research. We have adhered to standard research practices and have no conflicts of interest to declare.

## REPRODUCIBILITY STATEMENT

We have made efforts to ensure the reproducibility of our work. Complete experimental details are provided in Table 2 and Table 3 in the appendix, including all hyperparameters for both supervised fine-tuning and reinforcement learning phases. Our use of existing datasets (OpenR1-Math-220K for SFT and MetaMathQA for RL training) is fully disclosed, and the revision prompt used for trace rewriting is provided in Section I. The evaluation methodology and metrics for all four mathematical benchmarks (GSM8K, MATH500, AMC, and Minerva) are clearly specified. All algorithmic components of IGPO are detailed in Algorithm 1 with mathematical formulations provided throughout. We commit to releasing our implementation code upon publication to facilitate reproduction of our results.

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

## A USE OF LARGE LANGUAGE MODELS DISCLOSURE

LLMs were used only for minor editing (grammar and phrasing) and to generate speech narration for the supplementary presentation video from an author-written script. All research ideas, methods, experiments, analyses, and substantive writing were carried out by the authors without LLM assistance.

## B PRELIMINARIES

### B.1 MASKED DIFFUSION LARGE LANGUAGE MODELS

Masked diffusion LLMs (Austin et al., 2021; Sahoo et al., 2024; Shi et al., 2024; Ou et al., 2024; Lou et al., 2024) employ a forward diffusion process that progressively corrupts token sequences $x_0$ through introduction of $\mathtt{mask}$ tokens. This corruption process is parameterized by time $t \in [0, 1]$. At any given timestep $t$, the resulting sequence $x_t$ contains partial masking, where each token maintains a probability $\alpha_t$ of remaining unmasked. The noise schedule $\alpha_t$ exhibits strict monotonic decrease with respect to $t$. Complete masking occurs at $t = 1$, where all tokens in $x_1$ become masked. The training procedure for masked dLLMs follows a forward process through definition of $\alpha_t$ and a bidirectional unmasking predictor $f_\theta$ with learnable parameters. During each training step, we stochastically sample timestep $t \in [0, 1)$ and apply token masking according to the designated forward process. Given these corrupted sequences, the training objective seeks to recover the original tokens. The standard optimization criterion employs the negative evidence lower bound (NELBO), which provides an upper bound for the negative log-likelihood (NLL) of the training data. For masked dLLMs, this NELBO reduces to a weighted NLL formulation, with weighting coefficients derived from transformations of $\alpha_t$ (Sahoo et al., 2024, Equation (10)). For example, LLaDA (Nie et al., 2025) specifies the forward process through $\alpha_t = 1 - t$, yielding the following NELBO formulation:

$$-\mathbb{E}_{t\sim\mathcal{U}[0,1),\, x_0\sim p_{\text{data}},\, x_t\sim q_{t|0}(x_t|x_0)} \left[ \frac{1}{t} \sum_{k=1}^{|x_t|} \mathbb{1}[x_t^k = \mathtt{mask}] \log f_\theta(x_0^k \mid x_t) \right], \quad (6)$$

where $|x_t|$ denotes the sequence length of $x$, and $x^k$ represents the $k$-th token position. The loss computation is restricted to tokens masked at timestep $t$.

During prompt conditional generation, the model starts with a sequence where prompt tokens remain unmasked and continuation tokens are initially masked, then progressively unmasks the continuation tokens through ancestral sampling from the reverse process $p_\theta(x_s \mid x_t)$ for timesteps $t > s$, where the model $f_\theta$ provides the denoising predictions for masked positions. The reverse process maintains the property that unmasked tokens are carried over unchanged throughout all denoising steps.

### B.2 POLICY OPTIMIZATION FOR MASKED DIFFUSION LARGE LANGUAGE MODELS

Policy-gradient methods have gained widespread adoption for post-training LLMs (Ouyang et al., 2022; Bai et al., 2022; Li et al., 2023; Ahmadian et al., 2024). Online RL—particularly Group Relative Policy Optimization (GRPO)—has proved effective for improving language models (Shao et al., 2024; Guo et al., 2025; Team et al., 2025). GRPO (Shao et al., 2024) offers a computationally efficient alternative to PPO (Schulman et al., 2017) by using group-based statistics for advantage estimation, avoiding separate value-function training.

The GRPO objective integrates clipping for stability and reverse KL regularization:

$$\mathcal{L}_{\text{GRPO}}(\theta) = \mathbb{E}_{\substack{q\sim\mathcal{D} \\ o_1,\dots,o_G\sim\pi_{\theta_{\text{old}}}(\cdot|q)}} \left[ \frac{1}{G}\sum_{i=1}^{G}\frac{1}{|o_i|}\sum_{k=1}^{|o_i|} \min\left(\rho_i^k A_i, \text{clip}\left(\rho_i^k, 1-\varepsilon, 1+\varepsilon\right)A_i\right) - \beta D_{\text{KL}}\left[\pi_\theta(\cdot|q)\|\pi_{\text{ref}}(\cdot|q)\right] \right], \quad (7)$$

where $\rho_i^k = \frac{\pi_\theta(o_i^k|q,o_i^{<k})}{\pi_{\theta_{\text{old}}}(o_i^k|q,o_i^{<k})}$ is the likelihood ratio.

For a query $q$, GRPO samples $G$ responses $\{o_1, \dots, o_G\}$ from the behavior policy $\pi_{\theta_{\text{old}}}$ and assigns a *single sequence-level* advantage per response. Following Liu et al. (2025b), we use the unnormalized group-relative advantage $A_i = r(o_i) - \frac{1}{G}\sum_{j=1}^{G} r(o_j)$, where $r$ is the reward function. This scalar $A_i$ is shared by *all tokens* in $o_i$ when forming the tokenwise objective.

**Applying Policy Gradient Methods to Diffusion LLMs**    Applying GRPO to dLLMs is nontrivial. The objective in Equation (7) requires (i) *token-level* probabilities for importance ratios and (ii) *sequence-level* probabilities for KL regularization. Autoregressive models provide per-token conditionals via sequential factorization, enabling one-pass sequence scoring by the chain rule: $\log \pi_{\text{AR}}(o \mid q) = \sum_{k=1}^{|o|} \log \pi_{\text{AR}}(o^k \mid q, o^{<k})$. Accordingly, the reverse-KL decomposes as

$$D_{\text{KL}}\big[\pi_\theta(\cdot \mid q) \,\big\|\, \pi_{\text{ref}}(\cdot \mid q)\big] = \mathbb{E}_{o \sim \pi_\theta(\cdot \mid q)} \left[ \sum_{k=1}^{|o|} \log \frac{\pi_\theta(o^k \mid q, o^{<k})}{\pi_{\text{ref}}(o^k \mid q, o^{<k})} \right]. \tag{8}$$

In contrast, dLLMs do not admit a sequential factorization of $\pi(o \mid q)$. dLLM's generation invokes the unmasking predictor $f_\theta$ across $M$ denoising steps, making $\pi_\theta$ a composition of $M$ mappings. Exact tokenwise probabilities would require marginalization over denoising trajectories and maintaining (and differentiating through) full denoising trajectories, which is computationally prohibitive. To address this, recent work develops efficient approximations for policy optimization in masked diffusion LLMs. DiffuGRPO (Zhao et al., 2025) employs a mean-field approximation that yields *single-pass* estimates of both token-level and sequence-level terms, replacing explicit multi-step unrolling with a single-sample Monte Carlo estimate. While this introduces bias relative to the exact diffusion policy, it provides a practical framework for GRPO-style optimization on dLLMs. In our method, we adopt the mean-field estimators of Zhao et al. (2025) to compute the token-level importance ratios $\rho_i^k$ and the reverse-KL term with one forward pass per policy.

## C    RELATED WORK

### C.1    DIFFUSION LANGUAGE MODELS

Diffusion language models was first explored through continuous approaches that map discrete text to continuous representations, including learned embeddings, sequence-to-sequence conditioning, and binary bit representations (Chen et al., 2022; Li et al., 2022; Gong et al., 2023). Recently, discrete diffusion language models have been scaled up significantly, with masked diffusion established as a specific instance of discrete diffusion (Austin et al., 2021; Sahoo et al., 2024; Shi et al., 2024; Ou et al., 2024; Nie et al., 2024). Notable developments include DiffuLLaMA (Gong et al., 2025a) and Dream (Ye et al., 2025), both adapted from pretrained autoregressive LLMs. LLaDA (Nie et al., 2025) represents a breakthrough as a masked diffusion LLM trained from scratch using full-attention, achieving performance comparable to similarly-sized autoregressive models. These approaches are predominantly based on masked modeling. Unlike these full-attention dLLMs, Block Diffusion (Arriola et al., 2025) introduced a hybrid approach that models sequences block-by-block while applying diffusion within each block, enabling flexible length generation and improved inference efficiency through kv-caching. Recent commercial models like Mercury (Inception Labs et al., 2025) and Gemini Diffusion (DeepMind, 2025) have demonstrated the practical viability of diffusion-based code generation, achieving performance comparable to leading autoregressive models while offering significantly faster inference. More recent works have introduced caching and parallel decoding algorithms (Wu et al., 2025; Liu et al., 2025a; Ma et al., 2025; Israel et al., 2025; Sahoo et al., 2025; Hu et al., 2025) that significantly improve inference efficiency for masked diffusion language models. In this work, we focus on full-attention masked dLLMs.

### C.2    REINFORCEMENT LEARNING FOR DIFFUSION LANGUAGE MODELS

Applying reinforcement learning to diffusion language models presents unique challenges compared to autoregressive models. The primary obstacle is the intractability of likelihood functions in diffusion models, which necessitates approximating response likelihoods for policy optimization. This requirement introduces computational overhead and potential bias, particularly when approximation errors occur in policy ratios used for importance sampling. d1 proposed diffu-GRPO (Zhao et al., 2025) which adopts an efficient approximation through mean-field approximation. MMaDA (Yang et al., 2025) and diffucoder's coupled-GRPO (Gong et al., 2025b) further improve the masking strategy in log probabilities estimation to achieve better learning efficiency. LLaDA 1.5 (Zhu et al., 2025) tackles the variance issues in ELBO-based likelihood estimates through preference optimization. Recently, wd1 (Tang et al., 2025) addresses these challenges by reformulating policy optimization as a weighted likelihood objective that eliminates the need for policy ratios. SDPO (Han et al., 2025)

decomposes the diffusion trajectory alignment problem into stepwise subproblems that align the posterior at each diffusion step. Our inpainting method can also be applicable to some of the above online RL methods.

Additionally, a closely related work in RL for AR LLMs is Prefix-RFT (Huang et al., 2025), which samples prefixes from demonstrations to guide online exploration, though this is limited to left-to-right sequential generation that does not leverage the bidirectional conditioning capabilities of diffusion LLMs.

## D LENGTH-ALIGNED SFT: SFT TRACE REVISION LENGTH DISTRIBUTION COMPARISON

As illustrated in Figure 6, the original OpenR1-Math-220K dataset exhibits substantial token length diversity, with reasoning traces extending beyond 10,000 tokens while LLaDA's maximum context length is only 4096 tokens. Naively applying SFT on this dataset would result in many truncated sequences, and even for samples within the 4096-token limit, significant distribution mismatch persists across training phases—we use 256 tokens for RL sampling and 512 tokens for evaluation. Our rewriting using LLaMA-4-Maverick successfully constrains all traces to under 1500 tokens, creating alignment between SFT training, RL sampling, and evaluation phases. Additionally, while reflective behavior has been found helpful for LLaDA in prior work (Zhao et al., 2025), the excessive repeated reflective patterns in the original dataset are unsuitable for its constrained generation space. The rewriting process eliminates this redundancy while preserving essential reasoning structure.

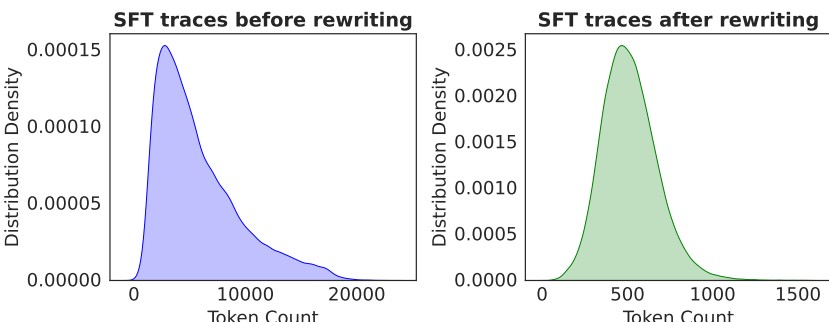

Figure 6: **Token Length Distribution of SFT Dataset Before and After Revision.** Comparison of token length distributions for the OpenR1-Math-220K dataset (94k math problems). After revision using LLaMA-4-Maverick, token lengths are constrained to below 1500 tokens, eliminating the extreme range of the original dataset where traces could exceed 20,000 tokens. This addresses the generation length mismatch across SFT training, RL sampling (256 tokens), and evaluation (512 tokens) phases.

# E   ABLATION: SFT ON HINT TRACES THEN APPLY GRPO VS IGPO

In our RL training setup, we assume access to ground-truth reasoning traces for every query in the training dataset. To investigate whether direct supervised fine-tuning on these traces provides comparable benefits to our elastic inpainting approach, we conduct an ablation study comparing two strategies: (1) applying SFT on the RL dataset's reasoning traces followed by standard GRPO sampling, versus (2) directly applying IGPO with elastic hint injection only when all generated responses are incorrect.

Specifically, we first fine-tune the LLaDA-8B-Instruct model on the MetaMath dataset's reasoning traces for 20 epochs, then apply standard GRPO sampling. We compare this against our IGPO approach, which selectively injects partial reasoning hints from the same MetaMath dataset only when zero-advantage scenarios occur (i.e., when all sampled responses yield incorrect rewards).

The results in Figure 7 demonstrate that IGPO consistently outperforms the SFT-first variant. Notably, after SFT on the MetaMath dataset for 20 epochs, the model's initial performance drops significantly compared to the original LLaDA-8B-Instruct baseline. This degradation occurs because the MetaMath dataset contains very concise reasoning traces, many shorter than our 256-token generation length limit. Consequently, the model adopts overly concise reasoning patterns that prove insufficient for the challenging problems in our evaluation benchmarks (such as AMC and Minerva).

While subsequent RL training can recover performance to some extent—as evidenced by the rapid improvement in early training steps—it ultimately fails to match the effectiveness of IGPO. This comparison highlights two key advantages of our approach: (1) the effectiveness of applying inpainting guidance selectively only when the model struggles with specific queries, rather than forcing a uniform reasoning style through SFT, and (2) the critical importance of reducing all-wrong group occurrences, which successfully recovers gradient signals from otherwise degenerate zero-advantage scenarios.

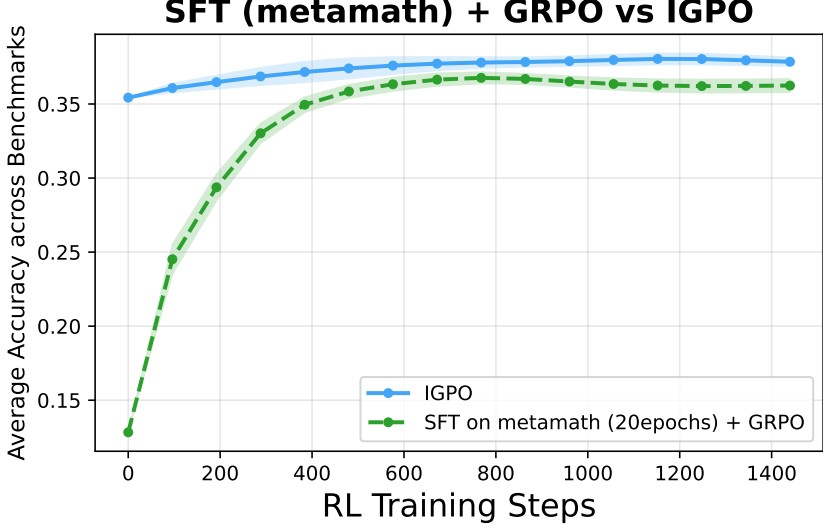

Figure 7: Comparison of SFT-first approach versus direct IGPO application. The SFT-first strategy involves fine-tuning on MetaMath reasoning traces for 20 epochs followed by standard GRPO, while IGPO applies inpainting-guided exploration elastically only during zero-advantage scenarios. IGPO demonstrates superior and more stable performance, avoiding the performance degradation caused by overly concise reasoning patterns learned during SFT on short traces. Results are averaged across four mathematical reasoning benchmarks with standard errors shown as shaded regions.

# F    EXPERIMENTS HYPERPARAMETERS

Table 2: Training Hyperparameters

| Parameter | Value |
|---|---:|
| **SFT Training Parameters** | |
| Per Device Train Batch Size | 4 |
| Hardware Configuration | 8×8 H100 GPUs |
| Gradient Accumulation Steps | 8 |
| Learning Rate | $5 \times 10^{-6}$ |
| LR Schedule | Warmup-stable-decay |
| LR Warmup Steps | 200 |
| LR Min Value | $1 \times 10^{-6}$ |
| LR Decay Period | Final 10% of steps |
| Number of Epochs | 100 |
| **RL Sampling Parameters** | |
| RL Online Sampling Generation Length $L$ | 256 |
| Diffusion Steps | 128 |
| Block Length | 32 |
| Sampling Temperature | 1.2 |
| Generations Per Group $G$ | 8 |
| **RL Training Parameters** | |
| Per Device Train Batch Size | 8 |
| Hardware Configuration | 8×8 H100 GPUs |
| Gradient Accumulation Steps | 1 |
| Effective Batch Size | 512 |
| KL Beta $\beta$ | 0.01 |
| Policy Gradient Inner Iterations per Generation $\mu$ | 4 |
| Learning Rate | $5 \times 10^{-7}$ |
| LR Schedule | Linear decay to 0 |
| LR Warmup Steps | 50 |
| LR Decay Period | 10 epochs |
| Training Steps | 1440 |
| Clip Ratio Epsilon $\varepsilon$ | 0.2 |
| **IGPO Specific Parameters** | |
| Chunk Size $|c_j| \sim \mathcal{U}[s_{\min}, s_{\max}]$ | $\mathcal{U}[5, 10]$ |
| Inpainting Ratio $\eta_i \sim \mathcal{U}[\eta_{\text{low}}, \eta_{\text{high}}]$ | $\mathcal{U}[0.2, 0.6]$ |
| replacement fraction $\lambda$ | 0.5 |
| Entropy-based Gradient Filtering for Inpainted Tokens $\tau$ | 0.2 |

### F.1 TEMPERATURE SELECTION FOR RL TRAINING

Following the methodology established by Polaris An et al. (2025) for scaling reinforcement learning on advanced reasoning models, we conduct a systematic analysis to determine the optimal sampling temperature for our RL training process. We evaluate our model's performance across different sampling temperatures by analyzing both Pass@5 and Average@5 scores on the MATH500 dataset. We also divide three temperature regions: low temperatures ($\leq 0.8$) yield high accuracy but reduced diversity in generated rollouts, restricting the model's ability to explore diverse reasoning paths; high temperatures ($\geq 1.6$) preserve rollout diversity but significantly degrade accuracy due to increased noise in token generation; and the middle Controlled Exploration Zone (0.9-1.5) provides the optimal trade-off between maintaining reasonable accuracy and achieving sufficient diversity for effective RL training. Based on this analysis, we select temperature $T = 1.2$ to balance exploration with sample quality and provide sufficient diversity for RL training.

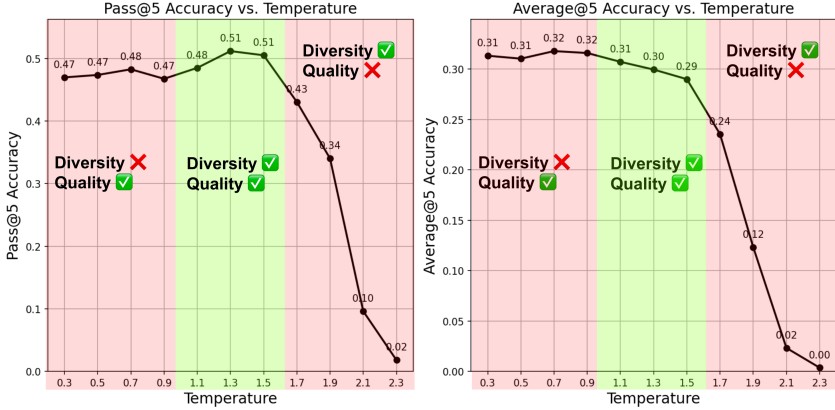

Figure 8: Performance analysis across different sampling temperatures on MATH500. The plot shows Pass@5 and Average@5 scores, revealing three distinct regions: low temperatures with high accuracy but low diversity, high temperatures with preserved diversity but degraded accuracy, and the middle region offering the desired trade-off. We select $T = 1.2$ for our RL training.

# G  INPAINTING GENERATION QUALITATIVE EXAMPLES

**Blue text:** Question Prompt    **Green text:** Injected Hints    **Black text:** Inpainted Generation

**Problem:** Circle $C$ has radius 6 cm. How many square centimeters are in the area of the largest possible inscribed triangle having one side as a diameter of circle $C$?

**Inpainting Input At Denosing Step 0:**

```
<|start_header_id|>user<|end_header_id|>
Respond in the following format:
<reasoning>
...
</reasoning>
<answer>
\ boxed{<Your answer>}
</answer>
Circle $C$ has radius 6 cm. How many square centimeters are in the area
of the largest possible inscribed triangle having one side as a diameter of
circle $C$?<|eot_id|><|start_header_id|>assistant<|end_header_id|>

<|mdm_mask|><|mdm_mask|>...<|mdm_mask|>
Since the diameter is twice the radius, the base of the triangle will have
length $2 \ times 6 = 12$ cm.
To <|mdm_mask|>...<|mdm_mask|> , we want the height to be as large
<|mdm_mask|>...<|mdm_mask|>
of the triangle is the distance from the midpoint of the base to the
circumference of the circle.
This is equal to the radius of the circle, <|mdm_mask|>...<|mdm_mask|>
<|mdm_mask|>...<|mdm_mask|> <|mdm_mask|>...<|mdm_mask|>
12 \ times <|mdm_mask|>...<|mdm_mask|>
```

**Inpainted Generation After Denoising:**

```
<reasoning>
We know to find the largest possible inscribed triangle in circle $C$, the base
of the triangle should be the diameter of the circle.
Since the diameter is twice the radius, the base of the triangle will have
length $2 \ times 6 = 12$ cm.
To find the largest possible area of the triangle, we want the height to be
as large as possible.
The largest possible height of the triangle is the distance from the midpoint of
the base to the
circumference of the circle.
This is equal to the radius of the circle, which is 6 cm.
Therefore, the area of the largest possible inscribed triangle is given by the
formula for the area of a triangle, which is $$\ frac{1}{2} \ times \ textbase
\ times \ textheight = \ frac{1}{2} \ times 12 \ times 6 = 36$$ square centimeters.
</reasoning>
<answer>
\ boxed{36}
</answer>
```

## H  EVALUATION HYPERPARAMETERS

We evaluate all models with a fixed decoding setup per benchmark. Unless otherwise noted, the same prompts and decoding hyperparameters are applied to both LLaDA-8B-Instruct and LLaDA-1.5.

Table 3: Evaluation hyperparameters by benchmark.

| Benchmark | Temp. | Block Len. | Diffusion Steps | Gen. Len. | Metric |
|---|---|---|---|---|---|
| GSM8K | 0.0 | 32 | 512 | 512 | pass@1 |
| MATH500 | 0.0 | 32 | 512 | 512 | pass@1 |
| AMC | 0.5 | 128 | 512 | 512 | avg@16 |
| Minerva | 0.0 | 128 | 512 | 512 | pass@1 |

## I  PROMPT FOR SFT TRACES REVISION

---

**Prompt for SFT traces revision sent to LLaMA 4 Maverick**

Please rewrite the original solution to make it more concise and easier to understand without changing the details. Please put the explanation in the solution between `<reasoning>` and `</reasoning>` and put the final numerical answer between `<answer>` and `</answer>` in boxed format. Please shorten or rewrite the rewritten solution to a random length between 100 and 1000 words while keeping sufficient details of the reasoning steps. Please do not return anything other than the rewritten solution.
**Example:**

```
<reasoning>
xxx
</reasoning>

<answer>
\boxed{14}
</answer>
```

**Original solution:** {generations}.
**Your response:**

---

## J    ROBUSTNESS OF IGPO TO NOISY REASONING TRACES

We evaluate IGPO's robustness when the ground-truth reasoning traces used for hint injection are corrupted with simulated realistic noise: we design numeric errors (e.g., "$8 \times 15 = 130$"), operator swaps (e.g., $\div \to \times$), logical word inconsistencies (e.g., "therefore" $\to$ "however"), and replacing hesitation tokens (e.g., "umm", "wait"). We vary the corruption rate $\rho$ from 0 to 0.5, where each token is independently corrupted with probability $\rho$. Examples of trace corruption is shown in Table 4.

Our evaluation results are shown in Figure 9 shows that IGPO remains effective even under introduced noise: although performance gradually degrades with noises, IGPO consistently outperforms GRPO. This robustness arises from IGPO's gating mechanisms: (i) only *partial* reasoning chunks are injected and the final answers must still be generated by the model, so only correctness-verified responses are used; and (ii) only the top 20% most uncertain injected tokens are allowed to contribute to the gradient, limiting exposure to incorrect hints.

We note that extremely corrupted traces violate the intended use of IGPO, whose goal is to leverage mostly correct reasoning to guide exploration; in practice, IGPO should not be applied to heavily inaccurate reasoning datasets.

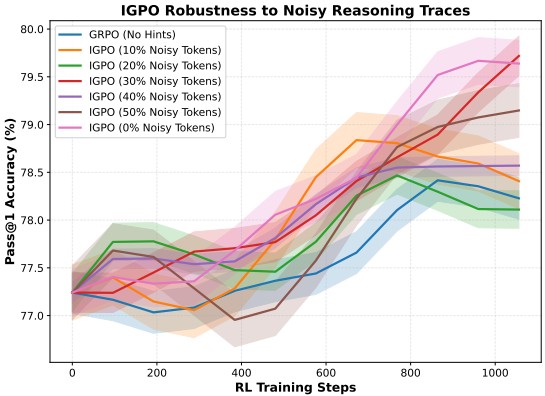

Figure 9: Pass@1 accuracy on GSM8K with generation length 256 under simulated reasoning-trace noise. A fraction $\rho$ of inpainting tokens are randomly corrupted.

| Type | Example |
|---|---|
| Original | Let's calculate: $8 \times 15 = 120$, then $120 \div 4 = 30$. Therefore, the answer is 30. |
| Number | Let's calculate: $8 \times 15 = 130$, then $130 \div 4 = 32$. Therefore, the answer is 32. |
| Operator | Let's calculate: $8 \times 15 = 120$, then $120 \times 4 = 480$. Therefore, the answer is 480. |
| Logic | Let's calculate: $8 \times 15 = 120$, then $120 \div 4 = 30$. However, the answer is 30. |
| Mixed | Umm let's calculate: $8 \times 15 = 125$, then $125 \div 4 = 31$. Wait, therefore the answer is 31. |

Table 4: Example reasoning traces corrupted. Red text indicates corrupted tokens.

# K  THEORETICAL ANALYSIS: GRADIENT RECOVERY AND KL-CONTROLLED INPAINTING

We analyze why IGPO improves policy optimization exactly in the regime where vanilla GRPO fails: the *all-wrong, zero-advantage* case. This is precisely the event that triggers inpainting in our algorithm. Throughout, $x \in \mathcal{X}$ denotes a query, $o \in \mathcal{O}$ a response sequence, and $r(o) \in \{0, 1\}$ a verifiable reward. The policy $\pi_\theta$ is parameterized by $\theta$.

Our main conclusions are: (1) In all-wrong groups, GRPO has a zero gradient, whereas IGPO restores a non-zero gradient whose magnitude scales as $\rho(1 - \rho)$, where $\rho$ is the *effective replacement ratio* of inpainted correct responses. (2) The same replacement mechanism admits a mixture-policy view: replacing an $\alpha$ fraction of the response distribution with the correct solution $o^\star$ yields a mixture policy $\pi_\alpha$ that weakly improves the expected reward and strictly improves it whenever the base policy is imperfect. (3) Partial hint injection controls the KL shift to the current policy linearly in the mixture weight $\alpha$, providing a soft trust-region-like stability guarantee. In our implementation, the batch-level replacement hyperparameter $\lambda$ upper-bounds the effective mixture weight $\alpha \approx \rho \leq \lambda$.

## K.1  SETUP AND ZERO-ADVANTAGE DILEMMA

We aim to maximize the expected reward

$$J(\theta) = \mathbb{E}_x\, J(\pi_\theta; x), \qquad J(\pi_\theta; x) = \mathbb{E}_{o \sim \pi_\theta(\cdot|x)}[r(o)]. \tag{9}$$

In GRPO, for each query $x$ we sample a group $\mathcal{S} = \{o_1, \ldots, o_G\}$ i.i.d. from $\pi_\theta(\cdot \mid x)$ and compute group-normalized advantages

$$\bar{r} = \tfrac{1}{G} \sum_{i=1}^{G} r(o_i), \qquad A_i = r(o_i) - \bar{r}, \tag{10}$$

which yields the per-query policy gradient estimator

$$\hat{g}_{\text{GRPO}}(x) = \frac{1}{G} \sum_{i=1}^{G} A_i \, \nabla_\theta \log \pi_\theta(o_i \mid x). \tag{11}$$

We focus on the *all-wrong* event

$$\mathcal{E}_{\text{wrong}} = \big\{ r(o_i) = 0 \ \forall i \in \{1, \ldots, G\} \big\}. \tag{12}$$

**Zero-gradient dilemma of GRPO.**  Conditioned on $\mathcal{E}_{\text{wrong}}$, we have $\hat{g}_{\text{GRPO}}(x) = \mathbf{0}$. This follows immediately since under $\mathcal{E}_{\text{wrong}}$, $r(o_i) = 0$ for all $i$, so $\bar{r} = \frac{1}{G} \sum_i r(o_i) = 0$ and $A_i = r(o_i) - \bar{r} = 0$ for all $i$. Substituting into equation 11 yields

$$\hat{g}_{\text{GRPO}}(x) = \frac{1}{G} \sum_{i=1}^{G} 0 \cdot \nabla_\theta \log \pi_\theta(o_i \mid x) = \mathbf{0}. \tag{13}$$

## K.2  GRADIENT RECOVERY AND POLICY IMPROVEMENT

We now analyze IGPO under $\mathcal{E}_{\text{wrong}}$. For a fixed query $x$, the IGPO batch construction proceeds as follows:

- Sample $G$ original responses $\{o_1, \ldots, o_G\}$ with $r(o_i) = 0$.

- Generate an additional set $\{\tilde{o}_1, \ldots, \tilde{o}_G\}$ via inpainting using hint injection based on the ground-truth trace $o^\star$.

- Keep only inpainted responses that pass correctness verification, $r(\tilde{o}_j) = 1$, and replace at most $\lfloor \lambda G \rfloor$ of the original failures, where $\lambda \in (0, 1)$ is a hyperparameter controlling the *maximum replacement fraction*.

Let $K$ be the number of inpainted correct responses used for replacement ($K \leq \lfloor \lambda G \rfloor$) and define the *effective replacement ratio*

$$\rho = \frac{K}{G} \in [0, \lambda]. \tag{14}$$

The augmented group is

$$\mathcal{S}' = \{o'_1, \ldots, o'_G\}, \quad |\{i : r(o'_i) = 1\}| = K, \quad |\{i : r(o'_i) = 0\}| = G - K. \tag{15}$$

Let

$$\mathcal{I}_{\text{correct}} = \{i : r(o'_i) = 1\}, \qquad \mathcal{I}_{\text{wrong}} = \{i : r(o'_i) = 0\}. \tag{16}$$

The new group-average reward is

$$\bar{r}_{\text{new}} = \frac{1}{G} \sum_{i=1}^{G} r(o'_i) = \frac{K}{G} = \rho, \tag{17}$$

and the corresponding advantages are

$$A'_i = \begin{cases} 1 - \rho, & i \in \mathcal{I}_{\text{correct}}, \\ -\rho, & i \in \mathcal{I}_{\text{wrong}}. \end{cases} \tag{18}$$

The per-query IGPO gradient estimator (omitting clipping and KL terms for clarity) is

$$\hat{g}_{\text{IGPO}}(x) = \frac{1}{G} \sum_{i=1}^{G} A'_i \nabla_\theta \log \pi_\theta(o'_i \mid x). \tag{19}$$

**Mixture policy interpretation.** The effective replacement ratio $\rho = K/G$ can be viewed as the weight of an idealized mixture policy that puts mass on the correct solution. For each query $x$ with ground-truth solution $o^\star$, define

$$\pi_\alpha(\cdot \mid x) = (1 - \alpha) \pi_\theta(\cdot \mid x) + \alpha \, \delta_{o^\star}, \qquad \alpha \in [0, 1], \tag{20}$$

where $\delta_{o^\star}$ is the Dirac distribution at $o^\star$. In this view, replacing $K$ of the $G$ samples by $o^\star$ corresponds to sampling approximately from $\pi_\alpha$ with $\alpha \approx \rho$; our implementation enforces $\rho \leq \lambda$ via the cap $K \leq \lfloor \lambda G \rfloor$.

We next relate this mixture to reward improvement. For a reference policy $\pi_\theta$, define the advantage

$$A^{\pi_\theta}(x, o) = r(o) - J(\pi_\theta; x). \tag{21}$$

In our single-step, verifiable-reward setting, the performance-difference lemma (Kakade and Langford, 2002) simplifies to

$$J(\pi') - J(\pi_\theta) = \mathbb{E}_x \, \mathbb{E}_{o \sim \pi'(\cdot \mid x)} \left[ A^{\pi_\theta}(x, o) \right], \tag{22}$$

for any comparison policy $\pi'$. For $\pi_\alpha$ in equation 20 and fixed $x$,

$$\begin{aligned} \mathbb{E}_{o \sim \pi_\alpha(\cdot \mid x)} \left[ A^{\pi_\theta}(x, o) \right] &= (1 - \alpha) \mathbb{E}_{o \sim \pi_\theta(\cdot \mid x)} \left[ A^{\pi_\theta}(x, o) \right] + \alpha \, A^{\pi_\theta}(x, o^\star) \\ &= (1 - \alpha) \cdot 0 + \alpha \big( r(o^\star) - J(\pi_\theta; x) \big) \\ &= \alpha \big( 1 - J(\pi_\theta; x) \big), \end{aligned} \tag{23}$$

since $r(o^\star) = 1$ and $\mathbb{E}_{o \sim \pi_\theta}[A^{\pi_\theta}(x, o)] = 0$ by definition in all-wrong case. Plugging into equation 22 yields

$$J(\pi_\alpha) - J(\pi_\theta) = \alpha \, \mathbb{E}_x \big[ 1 - J(\pi_\theta; x) \big]. \tag{24}$$

**Lemma 1** (Closed-form expected IGPO gradient). *Define the gradient expectations under the correct and wrong distributions separately:*

$$g_{correct}(x) = \mathbb{E}_{o' \sim \delta_{o^\star}} \left[ \nabla_\theta \log \pi_\theta(o' \mid x) \right] = \nabla_\theta \log \pi_\theta(o^\star \mid x), \tag{25}$$

$$g_{wrong}(x) = \mathbb{E}_{o' \sim \pi_\theta(\cdot \mid x)} \left[ \nabla_\theta \log \pi_\theta(o' \mid x) \right] \tag{26}$$

*where we note that $r(o') = 0$ for $o' \sim \pi_\theta(\cdot \mid x)$. Then*

$$g_{\text{IGPO}}(x) = \mathbb{E} \big[ \hat{g}_{\text{IGPO}}(x) \big] = \rho(1 - \rho) \big( g_{correct}(x) - g_{wrong}(x) \big). \tag{27}$$

*Proof.* The expectation in $g_{\text{IGPO}}(x) = \mathbb{E}[\hat{g}_{\text{IGPO}}(x)]$ is taken over all possible sets $\mathcal{S}' = \{o'_1, \ldots, o'_G\}$ such that: (1) the set has size $G$, and (2) exactly $K$ of the $G$ outputs are correct (sampled from $\delta_{o^\star}$) while the remaining $G - K$ are wrong (sampled from $\pi_\theta(\cdot \mid x)$ conditioned on being incorrect).

To compute this expectation, we first observe that for each valid set $\mathcal{S}'$, we can consider all $G!$ permutations of its elements. By symmetry, each position $i \in \{1, \ldots, G\}$ has the same marginal distribution when averaged over all permutations of all valid sets. Specifically, each $o'_i$ is drawn from the mixture distribution:

$$o'_i \sim \pi_\alpha(\cdot \mid x) \quad \text{with} \quad \alpha = \rho = \frac{K}{G}. \tag{28}$$

Therefore, we can decompose the expectation over groups into a sum of $G$ identical expectations, each over the mixture distribution $\pi_\alpha$:

$$
\begin{aligned}
g_{\text{IGPO}}(x) &= \mathbb{E}_{\mathcal{S}'} \left[ \frac{1}{G} \sum_{i=1}^{G} A'_i \nabla_\theta \log \pi_\theta(o'_i \mid x) \right] \\
&= \frac{1}{G} \sum_{i=1}^{G} \mathbb{E}_{o'_i \sim \pi_\alpha(\cdot \mid x)} \left[ A'_i \nabla_\theta \log \pi_\theta(o'_i \mid x) \right] \\
&= \mathbb{E}_{o' \sim \pi_\alpha(\cdot \mid x)} \left[ A'_i \nabla_\theta \log \pi_\theta(o' \mid x) \right], \tag{29}
\end{aligned}
$$

where the second equality follows from the symmetry argument and the third uses the fact that all $G$ terms are identical.

Now, expanding the mixture distribution $\pi_\alpha = (1 - \alpha)\pi_\theta + \alpha \delta_{o^\star}$ with $\alpha = \rho$:

$$
\begin{aligned}
&\mathbb{E}_{o' \sim \pi_\alpha(\cdot \mid x)} \left[ A'_i \nabla_\theta \log \pi_\theta(o' \mid x) \right] \\
&= (1 - \rho)\mathbb{E}_{o' \sim \pi_\theta(\cdot \mid x)} \left[ A'_i \nabla_\theta \log \pi_\theta(o' \mid x) \right] \\
&\quad + \rho \, \mathbb{E}_{o' \sim \delta_{o^\star}} \left[ A'_i \nabla_\theta \log \pi_\theta(o' \mid x) \right]. \tag{30}
\end{aligned}
$$

For outputs sampled from $\pi_\theta$ (conditioned on being wrong), we have $r(o') = 0$, so from equation 18, $A'_i = -\rho$:

$$
\begin{aligned}
(1 - \rho)\mathbb{E}_{o' \sim \pi_\theta | r(o')=0} \left[ A'_i \nabla_\theta \log \pi_\theta(o' \mid x) \right] &= (1 - \rho) \cdot (-\rho)\mathbb{E}_{o' \sim \pi_\theta | r(o')=0} \left[ \nabla_\theta \log \pi_\theta(o' \mid x) \right] \\
&= -\rho(1 - \rho) \, g_{\text{wrong}}(x). \tag{31}
\end{aligned}
$$

For outputs sampled from $\delta_{o^\star}$, we have $r(o') = 1$, so $A'_i = 1 - \rho$:

$$
\begin{aligned}
\rho \, \mathbb{E}_{o' \sim \delta_{o^\star}} \left[ A'_i \nabla_\theta \log \pi_\theta(o' \mid x) \right] &= \rho \cdot (1 - \rho)\mathbb{E}_{o' \sim \delta_{o^\star}} \left[ \nabla_\theta \log \pi_\theta(o' \mid x) \right] \\
&= \rho(1 - \rho) \, g_{\text{correct}}(x). \tag{32}
\end{aligned}
$$

Combining both terms:

$$
\begin{aligned}
g_{\text{IGPO}}(x) &= \rho(1 - \rho) \, g_{\text{correct}}(x) - \rho(1 - \rho) \, g_{\text{wrong}}(x) \\
&= \rho(1 - \rho) \big( g_{\text{correct}}(x) - g_{\text{wrong}}(x) \big). \tag{33}
\end{aligned}
$$

$\square$

**Theorem 1** (Gradient recovery and policy improvement). *Conditioned on $\mathcal{E}_{\text{wrong}}$ and for $0 < \rho < 1$, the expected IGPO gradient satisfies*

$$g_{\text{IGPO}}(x) = \rho(1 - \rho) \big( g_{\text{correct}}(x) - g_{\text{wrong}}(x) \big), \tag{34}$$

*and is non-zero whenever $g_{\text{correct}}(x) \neq g_{\text{wrong}}(x)$. Furthermore, for the mixture policy $\pi_\alpha$ in equation 20,*

$$J(\pi_\alpha) - J(\pi_\theta) = \alpha \, \mathbb{E}_x \big[ 1 - J(\pi_\theta; x) \big] \geq 0, \tag{35}$$

*with strict inequality whenever there exists a query $x$ such that $J(\pi_\theta; x) < 1$. The scalar factor $\rho(1 - \rho)$ governing the gradient magnitude is maximized at $\rho = 1/2$.*

*Proof.* The expression and non-degeneracy of $g_{\text{IGPO}}(x)$ follow directly from Lemma 1. Equation equation 24 implies $J(\pi_\alpha) - J(\pi_\theta) \geq 0$ since $0 \leq J(\pi_\theta; x) \leq 1$ for all $x$ (as $r(o) \in \{0, 1\}$). If there exists at least one $x$ with $J(\pi_\theta; x) < 1$, then $1 - J(\pi_\theta; x) > 0$ on a set of positive measure, so for any $\alpha > 0$ the improvement is strictly positive. Finally, the quadratic $f(\rho) = \rho(1 - \rho)$ has derivative $f'(\rho) = 1 - 2\rho$ and second derivative $f''(\rho) = -2 < 0$, so $f$ is maximized at $\rho = 1/2$. $\square$

**Connection to implementation.** In our method section, the *hint injection ratio* $\eta$ controls how many ground-truth chunks are injected during inpainting and thus only affects how candidate inpainted responses $\{\tilde{o}_i\}$ are generated. The *replacement hyperparameter* $\lambda$ caps how many correctness-verified inpainted responses enter the RL group: $K \leq \lfloor \lambda G \rfloor$, so the effective replacement ratio satisfies $\rho = K/G \leq \lambda$. In the mixture-policy view, the idealized mixture weight $\alpha$ is therefore realized in practice as a random effective weight $\alpha \approx \rho$ bounded by $\lambda$.

### K.3 KL Control via Partial Hint Injection

We now quantify the distributional shift induced by inpainting to argue that partial replacement leads to a controlled KL change relative to the current policy.

**Theorem 2** (KL control via partial hint injection). *For the mixture policy $\pi_\alpha$ in equation 20, the KL divergence to the current policy satisfies*

$$D_{\mathrm{KL}}\big(\pi_\alpha(\cdot \mid x) \,\|\, \pi_\theta(\cdot \mid x)\big) \;\leq\; \alpha \, D_{\mathrm{KL}}\big(\delta_{o^\star} \,\|\, \pi_\theta(\cdot \mid x)\big) = -\alpha \log \pi_\theta(o^\star \mid x). \tag{36}$$

*Proof.* The KL divergence $D_{\mathrm{KL}}(P\|Q)$ is convex in its first argument. For any distributions $P_1, P_2$ and $\alpha \in [0, 1]$,

$$D_{\mathrm{KL}}\big((1-\alpha)P_1 + \alpha P_2 \,\|\, Q\big) \leq (1-\alpha)D_{\mathrm{KL}}(P_1\|Q) + \alpha D_{\mathrm{KL}}(P_2\|Q). \tag{37}$$

Apply this with $P_1 = \pi_\theta(\cdot \mid x)$, $P_2 = \delta_{o^\star}$, $Q = \pi_\theta(\cdot \mid x)$:

$$\begin{aligned}
D_{\mathrm{KL}}\big(\pi_\alpha(\cdot \mid x) \,\|\, \pi_\theta(\cdot \mid x)\big) &\leq (1-\alpha)D_{\mathrm{KL}}\big(\pi_\theta(\cdot \mid x) \,\|\, \pi_\theta(\cdot \mid x)\big) + \alpha D_{\mathrm{KL}}\big(\delta_{o^\star} \,\|\, \pi_\theta(\cdot \mid x)\big) \\
&= \alpha D_{\mathrm{KL}}\big(\delta_{o^\star} \,\|\, \pi_\theta(\cdot \mid x)\big),
\end{aligned} \tag{38}$$

since $D_{\mathrm{KL}}(\pi_\theta\|\pi_\theta) = 0$. For the Dirac distribution,

$$D_{\mathrm{KL}}\big(\delta_{o^\star} \,\|\, \pi_\theta(\cdot \mid x)\big) = \sum_o \delta_{o^\star}(o) \log \frac{\delta_{o^\star}(o)}{\pi_\theta(o \mid x)} = \log \frac{1}{\pi_\theta(o^\star \mid x)} = -\log \pi_\theta(o^\star \mid x), \tag{39}$$

which proves equation 36. $\square$

**Interpretation and link to $\lambda$ and $\eta$.** When $\pi_\theta(o^\star \mid x)$ is small (the model is far from the correct solution), the full Dirac update ($\alpha = 1$) induces a large KL shift and may destabilize training. The bound equation 36 shows that using a mixture weight $\alpha < 1$ scales the KL divergence linearly with $\alpha$, implementing a soft trust-region–like constraint: IGPO injects strong supervised-like corrections while keeping the updated policy within a controlled KL neighborhood of the current iterate. Empirically, we observe that intermediate values of $\lambda$—corresponding to partial hint injection and hence intermediate $\rho$—yield the best performance.

## L   ABLATION: A SAMPLE-MATCHED GRPO BASELINE WITHOUT INPAINTING

To determine whether IGPO's improvements arise from its hint-guided exploration or merely from an increased number of generated trajectories, we compare against a GRPO baseline that matches IGPO's total sampling budget.

In GRPO + resample, whenever a GRPO group produces only incorrect responses, we resample additional trajectories from the current policy, matching the number of samples that IGPO would have generated. In GRPO + resample, all trajectories remain fully on-policy, and no hint tokens are injected.

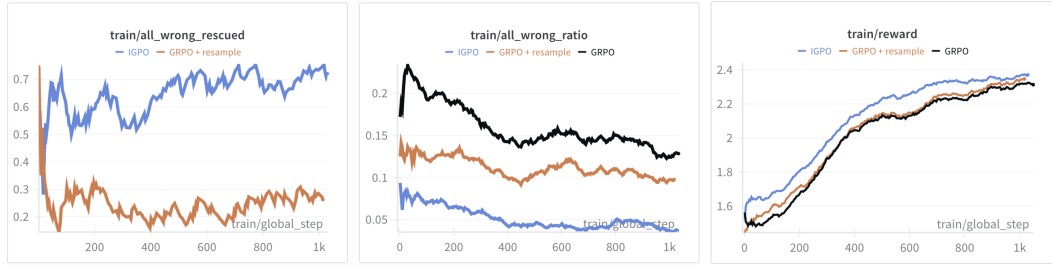

Figure 10: Training statistics comparing IGPO, standard GRPO, and GRPO with sample-matched resampling. We report the *all-wrong rescued* rate (the proportion of all-wrong groups in which resampling yields at least one correct response), the overall *all-wrong ratio*, and the training reward. IGPO achieves the lowest all-wrong ratio and the highest reward.

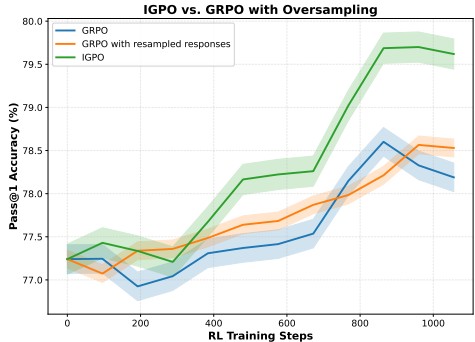

Figure 11: Evaluation curves on GSM8K (w/ generation length 256, greedy decoding), comparing IGPO, GRPO, and GRPO + resample. IGPO achieves higher pass@1, showing that its improvements are not explained by increased sampling alone.

As shown in Figure 10, despite using the same number of sampled trajectories, GRPO + resample consistently exhibits a higher all-wrong ratio and lower training rewards. Simply increasing the sampling budget does not resolve all-wrong cases, as the policy remains unable to discover correct solutions on its own. In contrast, IGPO substantially reduces the frequency of all-wrong groups through its hint-guided exploration mechanism. As shown in Figure 11, IGPO also outperforms both baselines in evaluation. These results confirm that IGPO's advantages stem from its inpainting-based guidance rather than from increased sampling volume. Partial hint injection provides targeted directional signals that pure resampling cannot replicate, while still preserving predominantly on-policy generation.

**SFT trace before rewritten by LLaMA 4 Maverick**
**Problem:** Find all primes between 1 and 15.

<think>
Okay, so I need to find all the prime numbers between 1 and 15. Hmm, let's start by remembering what a prime number is. From what I recall, a prime number is a natural number greater than 1 that has no positive divisors other than 1 and itself. So, numbers like 2, 3, 5, etc., are primes because you can't divide them evenly by any other number besides 1 and the number itself. On the other hand, numbers like 4, 6, 8, and so on are not primes because they can be divided by other numbers. For example, 4 can be divided by 2, so it's not prime. Alright, so the task is to list all primes between 1 and 15. Let me list out all numbers from 1 to 15 first to check them one by one. The numbers are: 1, 2, 3, 4, 5, 6, 7, 8, 9, 10, 11, 12, 13, 14, 15.

Now, let's go through each number and see if it's prime. Starting with 1. Wait, I remember that by definition, prime numbers are greater than 1. So 1 is not considered a prime. So we can skip 1.

Next up is 2. Let's check if 2 is prime. The divisors of 2 are 1 and 2. There are no other divisors, so yes, 2 is a prime number.

Moving on to 3. Similarly, the divisors are 1 and 3. No other numbers divide into 3, so 3 is prime.

Next is 4. Let's check divisors. 4 divided by 2 is 2, which is an integer, so 4 is not prime. It can be divided by 2, so it's composite.

Then 5. The divisors of 5 are 1 and 5. No other numbers divide into 5, so 5 is prime.

6. Checking divisors. 6 divided by 2 is 3, so it's divisible by 2 and 3. Not prime.

7. Divisors are 1 and 7. No other divisors, so 7 is prime.

8. Divided by 2 gives 4, so 8 is not prime.

9. Divided by 3 gives 3, so 9 is not prime (since 3 times 3 is 9).

10. Divided by 2 is 5, so 10 is not prime.

11. Let's see, 11. Divided by 2? 11 divided by 2 is 5.5, not an integer. Divided by 3? 11 divided by 3 is approximately 3.666... Not an integer. Divided by 4? 2.75, nope. 5? 2.2, nope. Since 11 is a prime number because none of these divisions result in integers, so 11 is prime.

12. Divided by 2 is 6, so composite.

13. Checking divisors. 13 divided by 2 is 6.5, 13 divided by 3 is about 4.333..., 4? 3.25, 5? 2.6, 6? 2.166... None of these are integers. So 13 is prime.

14. Divided by 2 is 7, so composite.

15. Divided by 3 is 5, so composite.

Let me recap the numbers between 1 and 15 and their primality:

1: Not prime
2: Prime
3: Prime
4: Not
5: Prime
6: Not
7: Prime
8: Not
9: Not
10: Not
11: Prime
12: Not
13: Prime
14: Not
15: Not

So compiling the primes from this list: 2, 3, 5, 7, 11, 13.

Wait a second, let me verify each one again just to be sure I didn't make a mistake.

Starting with 2: Definitely prime. It's the only even prime.

3: Prime, yes.

5: Prime, yes.

7: Prime. Correct.

11: Let's check again. 11 divided by 2 is not whole. 3? 3*3 is 9, 3*4 is 12. So no. 5? 5*2 is 10, 5*3 is 15. So 11 is prime.

13: Similarly, 13 divided by 2 is 6.5, 3 is 4.333, 5 is 2.6. So, prime.

Yes, those all check out. So primes between 1 and 15 are 2, 3, 5, 7, 11, and 13.

But hold on, sometimes people might make a mistake with 9. Wait, 9 is 3 times 3, so that's composite, which I had down. 15 is 3 times 5, composite. So all the non-primes are accounted for.

Is there any number I might have missed? Let's see. Let's list all numbers again:

2, 3, 5, 7, 11, 13. Are there any others?

Well, 1 is excluded. 4,6,8,9,10,12,14,15 are composite. So no, those six numbers are all primes between 1 and 15.

Wait, but the question says "between 1 and 15." Does that include 15? Hmm, the wording is "between 1 and 15," which could be a bit ambiguous. But usually, when someone says "between 1 and 15," they mean including both endpoints. However, since 15 is composite, it's excluded either way. So primes are 2,3,5,7,11,13.

Yes, that seems correct. I don't think I made any mistakes here. Let me cross-verify with another method, like the Sieve of Eratosthenes. The Sieve is a method where you list all numbers and cross out multiples.

Starting with 2, cross out multiples: 4,6,8,10,12,14.

Then 3, cross out multiples: 6,9,12,15.

Then 5, cross out 10,15.

7, cross out 14.

11 and 13 are primes, and their multiples are beyond 15. So remaining numbers are 2,3,5,7,11,13. Which matches our previous list.

Therefore, the primes between 1 and 15 are 2,3,5,7,11,13.

**Final Answer**

The primes between 1 and 15 are $\boxed{2}$, $\boxed{3}$, $\boxed{5}$, $\boxed{7}$, $\boxed{11}$, and $\boxed{13}$.

</think>

To find all the prime numbers between 1 and 15, we start by recalling that a prime number is a natural number greater than 1 that has no positive divisors other than 1 and itself. We then check each number from 1 to 15:

- **1**: Not a prime number (by definition).
- **2**: Prime (divisors: 1, 2).
- **3**: Prime (divisors: 1, 3).
- **4**: Not prime (divisible by 2).
- **5**: Prime (divisors: 1, 5).
- **6**: Not prime (divisible by 2, 3).
- **7**: Prime (divisors: 1, 7).
- **8**: Not prime (divisible by 2).
- **9**: Not prime (divisible by 3).
- **10**: Not prime (divisible by 2, 5).
- **11**: Prime (divisors: 1, 11).
- **12**: Not prime (divisible by 2, 3, 4, 6).
- **13**: Prime (divisors: 1, 13).
- **14**: Not prime (divisible by 2, 7).
- **15**: Not prime (divisible by 3, 5).

Using the Sieve of Eratosthenes method to cross out non-prime numbers confirms the primes listed above. Therefore, the prime numbers between 1 and 15 are:

$$\boxed{2}, \boxed{3}, \boxed{5}, \boxed{7}, \boxed{11}, \boxed{13}$$

**SFT trace after rewritten by LLaMA 4 Maverick**
**Problem:** Find all primes between 1 and 15.

<reasoning>
To find all the prime numbers between 1 and 15, we recall that a prime number is a natural number greater than 1 that has no positive divisors other than 1 and itself.
We then check each number from 1 to 15 to determine if it is prime.
- **1**: Not a prime number (by definition).
- **2**: Prime (divisors: 1, 2).
- **3**: Prime (divisors: 1, 3).
- **4**: Not prime (divisible by 2).
- **5**: Prime (divisors: 1, 5).
- **6**: Not prime (divisible by 2, 3).
- **7**: Prime (divisors: 1, 7).
- **8**: Not prime (divisible by 2).
- **9**: Not prime (divisible by 3).
- **10**: Not prime (divisible by 2, 5).
- **11**: Prime (divisors: 1, 11).
- **12**: Not prime (divisible by 2, 3, 4, 6).
- **13**: Prime (divisors: 1, 13).
- **14**: Not prime (divisible by 2, 7).
- **15**: Not prime (divisible by 3, 5).
The Sieve of Eratosthenes method confirms the primes listed above.
</reasoning>
<answer>
$\boxed{2, 3, 5, 7, 11, 13}$
</answer>

