# OpenReview forum: "Inpainting-Guided Policy Optimization for Diffusion Large Language Models"
_ICLR.cc/2026/Conference — ICLR 2026 Poster_

### Official Review · Reviewer_LgBR · 2025-10-26

**Soundness:** 3
**Presentation:** 3
**Contribution:** 3
**Rating:** 4
**Confidence:** 4

**Summary:**

This paper introduces IGPO (inpainting guided policy optimization) for optimizing diffusion language models with RL which uses ground truth correct trajectories in order to learn in questions where all of the generations are incorrect. To do this, the algorithm resamples more trajectories for questions where all generations are wrong, but inserting text from the correct ground truth trajectory to be inpainted around. Followed by a standard GRPO update on it. This provides "hints" for the model during generation. The authors show that this increase performance in math reasoning tasks.

**Strengths:**

- this paper is well written
- this paper considers an important problem, both improving overall performance and utilizing ground truth responses.
- the results of this paper are encouraging for the method.

**Weaknesses:**

- while it does seem to work, the entropy-based gradient filtering seems like a method in order to deal with the off policy-ness of the hint responses. Being able to deal with this off-policy would be desirable instead (although harder).
- the improvement is encouraging, but it is unclear whether it comes from increasing generation number or the hints (see questions)
- My overall feeling is this seems to improve performance slightly, but there is more work to be done to show that this is the *right* way to use gold trajectories. Ie. How does it compare to distillation methods?

**Questions:**

- how many of these new responses get clipped? I feel like clipping would null many of these since the trajectory would be off policy?
- I believe that the fair comparison would actually be to GRPO which oversampled on incorrect trajectories. Ie. Do hint tokens actually matter?
- does `without Inpaint` in Figure 4 mean that one adds oversampled correct trajectories into the batch as well?
- does SFT on the rewritten trajectories move it closer to on policy?

---

> ### Author Response · Authors · 2025-11-22
> **rebuttal reply (1/n)**
>
> Dear reviewer LgBR,
>
> We are thankful for the constructive feedback and we appreciate your recognition of the importance of our scope and our empirical results.
>
> Below we address your constructive feedbacks with clarifications and additional experimental details.
>
> > Weakness 1: while it does seem to work, the entropy-based gradient filtering seems like a method in order to deal with the off policy-ness of the hint responses. Being able to deal with this off-policy would be desirable instead (although harder). Q1: how many of these new responses get clipped? I feel like clipping would null many of these since the trajectory would be off policy?
>
> Yes — in our experiments, we observe that the **actual clip ratio remains near zero throughout training**. This shows that the off-policy nature of hint-injected tokens is effectively mitigated after our proposed entropy filtering and importance-ratio clipping, leading to stable training. To emphasize, IGPO uses two gates that aggressively regulate inpainting:
>
> 5. **Entropy filtering:** we retain only the top 20% most uncertain hint tokens (selected via ablation).
> 6. **Importance-ratio clipping (for all tokens):** across training, we observe that the clip ratios for both hint tokens and all tokens remain near zero, meaning that (1) all entropy-filtered hint tokens lie inside the clipped region and therefore contribute no unclipped gradient, and (2) even the model’s own infilled tokens stay close to their behavior policy, indicating that the overall update remains stable and well regularized.
>
> **We also want to note that the learning is still dominated by the model’s on-policy generations, where 80% of tokens are generated by the model,** while the hint tokens act only as a small guiding signal. Empirically we show IGPO consistently outperforms the GRPO baseline, which provides no learning signal at all in all-wrong cases.
>
> > Weakness 2: the improvement is encouraging, but it is unclear whether it comes from increasing generation number or the hints (see questions)  Q2: I believe that the fair comparison would actually be to GRPO which oversampled on incorrect trajectories. Ie. Do hint tokens actually matter?
>
> We appreciate the suggestion and conducted an ablation to isolate the effect of hint tokens from just re-sampling. Specifically, we implemented a *sample-matched* “GRPO \+ resample” baseline that, whenever a group is all-wrong, we draw additional on-policy rollouts matching IGPO’s total number of generated trajectories—without injecting any hint tokens.
>
> We provide a detailed analysis in Appendix L, including new figures reporting training reward, all-wrong rescue rate, and all-wrong group ratios (in Fig 10 and 11). We show that although GRPO resampling modestly lowers the all-wrong ratio, it remains weaker than IGPO in both training dynamics and final accuracy. This shows IGPO’s gains arise from inpainting with hints rather than from increased sampling compute. The table below summarizes the quantitative comparison. Please refer to the revised paper for detailed figures.
>
> | GSM8K result | GRPO | GRPO \+ resample | IGPO |
> | :---- | :---- | :---- | :---- |
> | pass@1 acc (at \~1k step) | 78.0 | 78.9 | **79.6** |
> | all-wrong ratio (at \~1k step) | \~13% | \~10% | **\~3%** |
>
>
>
> > Weakness 3: My overall feeling is this seems to improve performance slightly, but there is more work to be done to show that this is the right way to use gold trajectories. Ie. How does it compare to distillation methods?
>
> Thank you for the suggestion. We have conducted an experiment comparing IGPO against a distillation-style alternative by performing SFT on the gold reasoning traces (used in IGPO, not in prior SFT), as shown in Appendix E and Figure 7\. This SFT baseline corresponds to distilling the same trajectories that IGPO uses for inpainting. Our findings show that SFT on these inpainting traces (from Metamath) actually degrades performance: the traces are extremely concise compared to dense reasoning traces (e.g., OpenR1), and the model quickly overfits to these short patterns, leading to a large drop. Subsequent GRPO recovers performance but still underperforms IGPO. By contrast, **IGPO leverages the same gold trajectories more effectively by injecting *partial* hints only when the policy fails, avoiding distribution shift and reducing all-wrong groups**.
>
> As shown in the table below, elastic inpainting is a more robust and performant way to incorporate gold trajectories than distillation alone. While SFT on Metamath traces produces bad results, those same traces become beneficial when used as IGPO hints, showing **inpainting can transform even suboptimal gold trajectories into useful learning signals.**
>
> |  | LLaDA | LLada \+ IGPO | LLada \+ SFT | LLada \+SFT then GRPO |
> | :---- | :---- | :---- | :---- | :---- |
> | Average Accuracy across 4 Benchmarks | 0.353 | **0.372** | 0.121 | 0.359 |

---

> ### Author Response · Authors · 2025-11-22
> **rebuttal reply (2/n)**
>
> > Q3: does without Inpaint in Figure 4 mean that one adds oversampled correct trajectories into the batch as well?
>
> The “without Inpaint” curve in Figure 4 corresponds to standard GRPO and does not include any oversampled correct trajectories,
>
> > Q4: does SFT on the rewritten trajectories move it closer to on policy?
>
> In our experiments, we observe that the clip ratios for all tokens remain near zero both before and after applying length-aligned SFT, and the overall policy ratios are also similar across the two runs. We think SFT does not have large impact on the “on-policyness” because IGPO has two gates as described above that makes it not altering the “on-policyness” significantly. Because the number of inpainted (off-policy) tokens is small, and IGPO is only triggered during all-wrong case and even when triggered, up to **80% of the tokens are still generated by the model itself**. Therefore the model is mostly on-policy with its initialization checkpoint.
>
> We hope our new clarifications and additional experiments address your concerns. We are happy to address any further questions you might have.

---

> > ### Author Response · Authors · 2025-11-26
> > **gentle reminder**
> >
> > Dear reviewer LgBR,
> >
> > Thank you again for your constructive review. We have added new experiments including a sample-size-matched “GRPO + resample” baseline (Appendix L), RL run clipping statistics, and comparisons to SFT-based distillation baselines, and clarifications on the inpainting ablations. Please kindly let us know if we can address any additional concerns before the discussion period ends. Thank you!

---

> > > ### Comment · Reviewer_LgBR · 2025-11-26
> > >
> > > I thank the author for their response and the additional experiments conducted. I am encouraged by the results of the ablations and appendix J which partially deals with my off-policyness concern as well. As such, I am advocating to accept this paper.

---

> > > > ### Author Response · Authors · 2025-11-26
> > > > **thank you for raising score**
> > > >
> > > > Thank you for raising the score and suggesting the constructive ablations! We are glad that we addressed your concerns, and we will incorporate these ablations into the final version of the paper.

---

### Official Review · Reviewer_L2eU · 2025-11-01

**Soundness:** 2
**Presentation:** 3
**Contribution:** 2
**Rating:** 4
**Confidence:** 2

**Summary:**

The paper proposes Inpainting-Guided Policy Optimization (IGPO), a reinforcement learning framework designed to enhance exploration in diffusion-based large language models under sparse reward settings. The key idea is to inject partial ground-truth reasoning traces as inpainting hints into masked regions during RL sampling, allowing the model to receive guided feedback while retaining its own generative reasoning. This strategy effectively mitigates the zero-advantage problem in group-based policy optimization methods such as GRPO, where uniformly incorrect responses yield zero gradients and inefficient learning. Experiments on mathematical reasoning datasets demonstrate the effectiveness of the proposed method.

**Strengths:**

The proposed use of inpainting for guided exploration in RL for dLLMs is a creative and well-motivated exploitation of architecture-specific inductive bias.

The approach is validated with thorough quantitative results on four widely recognized mathematical reasoning benchmarks,substantial gains over both prior masked dLLM methods and non-diffusion LLM baselines are reported
Ablation studies are thorough.

**Weaknesses:**

The formulation of IGPO objective in the sampling procedure(Eq 5)  is difficult to follow.

What does "Advantages $A_{i}$ are computed normally" mean?

The theoretical analysis is weak, provides limited justification for the proposed inpainting mechanism.

**Questions:**

Could the authors provide theoretical analysis to justify why the proposed inpainting mechanism improves policy optimization performance?

How robust is the proposed trace rewriting method? Have the authors evaluated it against simpler rewriting heuristics or with different base models for inpainting?

For domains lacking high-quality reasoning traces, how does IGPO’s performance degrade or adapt under less informative supervision?

---

> ### Author Response · Authors · 2025-11-22
> **rebuttal reply (1/n)**
>
> Dear reviewer L2eU,
>
> We are thankful for your constructive feedback and for your acknowledgement that our algorithm as "creative exploitation of architecture-specific inductive bias." and our ablations as thorough.
>
> Below we address your concerns and questions with new revisions, experiments and theoretical results:
>
> > W1: The formulation of IGPO objective in the sampling procedure(Eq 5\) is difficult to follow.
>
> > W2: What does "Advantages  are computed normally" mean?
>
> Thank you for these helpful comments on clarity. We agree that the IGPO objective around Eq. 5 required additional clarification. In the revised pdf, we have rewritten the surrounding text to explicitly describe the augmented sampling process and to clearly distinguish IGPO from standard GRPO. As shown in the updated method section, **the only modification introduced by IGPO lies in the sampling step** and only when the all-wrong condition is detected, the original on-policy responses are partially replaced by K correctness-verified inpainted samples. The rest of the GRPO objective remains unchanged.
>
> Regarding “advantages are computed normally,” we modified the paper to state this explicitly in the paper: **IGPO uses the exact same advantage definition as GRPO**, given in Eq. 2, because IGPO only modifies the sampling process.
>
> Hope this clarifies the confusion. Please let us know if you have other confusions regarding the method.
>
> >W3: The theoretical analysis is weak, provides limited justification for the proposed inpainting mechanism.
> Q1: Could the authors provide theoretical analysis to justify why the proposed inpainting mechanism improves policy optimization performance?
>
> We appreciate the request for a stronger theoretical justification. In the revised version, we have added extensive theoretical analysis in **Appendix K**, where we prove two main theorems.
>
> * **Theorem 1: Gradient recovery and policy improvement.** We formally analyze the all-wrong, zero-advantage regime where vanilla GRPO yields an identically zero gradient and cannot learn. We show that IGPO’s inpainting restores a non-degenerate gradient whose magnitude depends on the fraction of inpainted correct responses, and that—under an idealized mixture-policy view—replacing a small fraction of the policy’s mass with the ground-truth solution strictly improves expected reward whenever the base policy is imperfect.
>
> * **Theorem 2: KL control via partial hint injection.** We then quantify the distribution shift induced by inpainting and prove that partial injection on hint tokens leads to a KL divergence that scales linearly with the injection fraction. This provides a soft trust-region–like guarantee: IGPO can inject strong supervised-like corrections while keeping each update within a controlled KL neighborhood of the current policy, which explains the stability we observe empirically.
>
> We believe these new theory results address your W3 and Q1 in that IGPO improves policy optimization by (i) restoring a meaningful learning signal exactly in the all-wrong case where GRPO provide no learning at all, and (ii) doing so via a KL-controlled partial hint injection mechanism gives provable reward improvement while maintaining training stability, which **matches with our experimental results in Figure 4 that partial injection is better than full hint injection.**
>
> > Q2: How robust is the proposed trace rewriting method? Have the authors evaluated it against simpler rewriting heuristics or with different base models for inpainting?
>
> * The trace rewriting is proposed for length-distribution aligned SFT training and it is not related to inpainting. For rewriting heuristic: we used a very concise and straightforward rewriting guiding prompt as shown in Appendix I, and it is very straightforward that we do not tune the prompt and it has shown to outperform original traces by a large margin as shown in our main results in Table 1\.
> * We also conducted a new experiment on using the same rewriting prompt on another reasoning datasets: Openthoughts-114k, and we show that our rewriting method still outperform vanilla SFT on the original traces by a large margin.
>
> |  | SFT on original openthoughts | SFT on rewritten openthoughts traces |
> | :---- | :---- | :---- |
> | Pass@5 Performance on GSM8K (after 1 epoch of training) | 90.1 | 94.5 |

---

> ### Author Response · Authors · 2025-11-22
> **rebuttal reply (2/n)**
>
> > Q3: For domains lacking high-quality reasoning traces, how does IGPO’s performance degrade or adapt under less informative supervision?
>
> We run new experiment to explicitly study robustness to imperfect reasoning traces. To evaluate IGPO under noisy hints, we systematically corrupt ground-truth reasoning traces used for hint injection whenever all-wrong groups are detected. We introduce realistic perturbations such as: (i) numeric mistakes, (ii) operator confusions, (iii) logical discourse errors, and (iv) hesitation fillers.
>
> Our evaluation results are shown in the revision pdf’s Appendix J, we show RL training curve in Figure 9\. Results shows that **IGPO remains effective even under introduced noise**: although performance gradually degrades with noises, IGPO consistently outperforms GRPO.
>
> We show a summarized fix step evaluation in the table below, and please refer to the revised paper for more detailed training curves:
> |  | GRPO | IGPO w/o noise | IGPO w/ 10% noise | IGPO w/ 20% noise | IGPO w/ 30% noise | IGPO w/ 40% noise | IGPO w/ 50% noise |
> | :---- | :---- | :---- | :---- | :---- | :---- | :---- | :---- |
> | Gsm8k evaluation  | 0.780 | 0.796  | 0.778 | 0.782 | 0.799 | 0.785 | 0.791 |
>
> This robustness arises from IGPO’s gating mechanisms: (i) only partial reasoning chunks are injected and the final answers must still be generated by the model, so only correctness-verified responses are used; and (ii) only the top 20% most uncertain injected tokens are allowed to contribute to the gradient, limiting exposure to incorrect hints.
>
> We note that extremely corrupted traces violate the intended use of IGPO, whose goal is to leverage mostly correct reasoning to guide exploration; in practice, IGPO should not be applied to heavily inaccurate reasoning datasets.
>
> We hope these new theoretical results, clarifications and additional experiments address your concerns. We are happy to address any further questions you might have.

---

> > ### Author Response · Authors · 2025-11-26
> > **gentle reminder**
> >
> > Dear reviewer L2eU,
> >
> > Thank you again for your helpful review. We have added theoretical analysis (Appendix K), robustness experiments on noisy traces (Appendix J), rewriting SFT experiment on Openthoughts, and clarifications to the IGPO objective and advantage computation. Please let us know if we can clarify anything further before the discussion period ends. Thank you!

---

### Official Review · Reviewer_raEu · 2025-11-01

**Soundness:** 2
**Presentation:** 3
**Contribution:** 2
**Rating:** 4
**Confidence:** 2

**Summary:**

The paper introduces IGPO, a reinforcement learning framework that leverages the inpainting capability of masked diffusion large language models to overcome exploration inefficiency in RL finetuning. Instead of relying on full solutions, IGPO injects partial ground-truth reasoning hints during generation, guiding exploration toward promising reasoning paths while trying to maintain on-policy learning. Combined with a Length-Aligned Supervised Fine-Tuning stage using concise rewritten reasoning traces and entropy-based gradient filtering for stability, IGPO achieves state-of-the-art performance on GSM8K, Math500, AMC, and Minerva, improving sample efficiency and robustness over baseline methods.

**Strengths:**

1. The paper is well-written and thoughtfully presented, making its core ideas clear and easy to follow.

1. The proposed algorithm elegantly exploits the unique bidirectional and inpainting capabilities of diffusion LLMs to form a guided exploration strategy that resolves the zero-advantage problem in RL finetuning.

2. It demonstrates clear empirical gains, achieving state-of-the-art results across multiple mathematical reasoning benchmarks.

**Weaknesses:**

1. The proposed method relies heavily on the unique characteristics of diffusion LLMs and is therefore not a general solution to the zero-advantage problem applicable to broader classes of language models.

2. Compared with standard GRPO, IGPO additionally requires the RL dataset to include ground-truth reasoning paths, which further limits its applicability and use scenarios.

3. The use of reasoning hints violates the on-policy requirement of underlying RL algorithms, potentially leading to bias in the optimization objective.

4. Using reasoning hints during the RL process may constrain the model's ability to explore freely, making the final performance dependent on the quality of the provided expert reasoning hints.

**Questions:**

1. IGPO assumes access to accurate ground-truth reasoning traces during RL. How robust is the method to imperfect or noisy reasoning annotations? Have the authors considered evaluating IGPO with reasoning traces of varying quality to assess its sensitivity?

2. Would it be possible to include a discussion or comparison between IGPO and more recent approaches addressing the zero-advantage problem, such as [1]?

[1] Le, Thanh-Long V., et al. "No prompt left behind: Exploiting zero-variance prompts in llm reinforcement learning via entropy-guided advantage shaping." arXiv preprint arXiv:2509.21880 (2025).

---

> ### Author Response · Authors · 2025-11-22
> **rebuttal reply (1/n)**
>
> Dear reviewer raEu,
>
> We are thankful for the constructive feedback and for your acknowledgement that our algorithm "elegantly exploits the unique bidirectional and inpainting capabilities of diffusion LLMs,"
>
> Below we address your concerns and questions with new experiments and theoretical results:
>
> > Weakness 1: The proposed method relies heavily on the unique characteristics of diffusion LLMs and is therefore not a general solution to the zero-advantage problem applicable to broader classes of language models.
>
> We would like to clarify the scope of our paper: Our intention is to design algorithms that make meaningful use of the unique capabilities of dLLMs.
>
> 1. In this context, **there is a growing body of RL work tailored specifically to masked diffusion LLMs**—such as D1 [1], DiffuCoder [2], wd1 [3], and MMaDA [4]—which similarly develop methods that leverage properties unique to dLLMs. IGPO follows this line of thinking by making principled use of inpainting, a capability that the you highlighted as a key strength of the paper in strength 2.
> 2. Additionally, the idea and the insights from our work also applies to autoregressive LLMs—combining supervised signals with RL to improve sample efficiency—is an active research direction for autoregressive LLMs as well. Recent work such as Prefix-RFT\[5\], UFT\[6\] pursue similar goals where they add ground truth prefix during RL sampling. Our insights from our extensive ablation studies such as the inpainting ratio and entropy filtering may inspire future work across architectures in this direction.
>
>
> \[1\] d1: Scaling reasoning in diffusion large language models via reinforcement learning. arXiv:2504.12216.
> \[2\] Diffucoder: Understanding and improving masked diffusion models for code generation. arXiv:2506.20639.
> \[3\] wd1: Weighted Policy Optimization for Reasoning in Diffusion Language Models. arXiv:2507.08838.
> \[4\] MMaDA: Multimodal large diffusion language models. arXiv:2505.15809.
> \[5\] Blending supervised and reinforcement fine-tuning with prefix sampling. arXiv:2507.01679.
> \[6\] UFT: Unifying supervised and reinforcement fine-tuning. arXiv:2505.16984.
>
> > Weakness 2: Compared with standard GRPO, IGPO additionally requires the RL dataset to include ground-truth reasoning paths, which further limits its applicability and use scenarios.
>
> We appreciate this concern and would like to clarify several key points:
>
> 1. **IGPO is designed to combine the strengths of SFT and RL.** Rather than positioning IGPO as a pure RL method, our goal is to leverage the complementary benefits of SFT and RL while avoiding their limitations for dLLMs. Pure RL is computationally expensive with poor sample efficiency in zero-advantage cases. SFT assumes access to reasoning datasets and can cause distribution shift/forgetting when training data doesn't match the policy's generation patterns. IGPO addresses both limitations by applying inpainting only when the model struggles (zero-advantage cases).
> 2. **IGPO does not require full reasoning traces for the entire RL dataset.** It only requires traces for problems the current model cannot solve, largely reducing the requirement.
> 3. Empirically, IGPO is a more robust way to incorporate gold trajectories. Please check out our ablation in Appendix E comparing pure SFT, IGPO, and SFT+GRPO show that IGPO's elastic guidace outperforms both uniform SFT (which degrades initial performance due to large distribution shift, as in Figure 7\) and standard GRPO. IGPO reduces all-wrong groups by approximately 60% (Figure 1b), translating to substantial gains in sample efficiency. As shown in the table below, elastic inpainting is a more robust and performant way to incorporate gold trajectories than SFT alone. While SFT on Metamath traces produces bad results, those same traces become beneficial when used as IGPO hints, showing **inpainting can transform even suboptimal gold trajectories into useful learning signals.**
>
> | Method | LLaDA | LLada \+ IGPO | LLada \+ SFT | LLada \+SFT then GRPO |
> | :---- | :---- | :---- | :---- | :---- |
> | Average Accuracy across 4 Benchmarks | 0.353 | **0.372** | 0.121 | 0.359 |

---

> ### Author Response · Authors · 2025-11-22
> **rebuttal reply (2/n)**
>
> > Weakness 3: The use of reasoning hints violates the on-policy requirement of underlying RL algorithms, potentially leading to bias in the optimization objective.
>
> We agree that IGPO introduces off-policyness on injected tokens, but in our experiments stats and our newly added extensive theoretical analysis in Appendix K that this deviation is well-controlled and is provably better than no learning in the all-wrong cases. The main reasons are:
>
> 1. **We show in our newly added theoretical results in Appendix K that IGPO updates are bounded by KL and strictly reward improving**  than no learning at all in GRPO in all wrong cases.
> 2. We observe empirically that the **actual importance ratio clip ratio remains near zero throughout training**. This shows that the off-policy nature of hint-injected tokens is effectively mitigated after our proposed entropy filtering and importance-ratio clipping, leading to stable training.
>
>     To emphasize, IGPO uses two gates that aggressively regulate inpainting:
>    - Entropy filtering: we retain only the top 20% most uncertain hint tokens (selected via ablation).
>    - Importance-ratio clipping (for all tokens)**:** across training, we observe that the clip ratios for both hint tokens and all tokens remain near zero, meaning that (1) all entropy-filtered hint tokens lie inside the clipped region and therefore contribute no unclipped gradient, and (2) even the model’s own infilled tokens stay close to their behavior policy, indicating that the overall update remains stable and well regularized.
> 3. The learning is still dominated by the model’s on-policy generations, where **80% of tokens are still generated by the model,** while the hint tokens act only as a small guiding signal. Empirically we show IGPO consistently outperforms the GRPO baseline, which provides no learning signal at all in all-wrong cases.
> 4. Furthermore, Appendix E shows that a truly off-policy alternative—naively applying SFT on ground-truth reasoning traces followed by GRPO—can hurt performance, whereas IGPO’s **selective, partial** hint injection avoids this degradation.
>
> > Weakness 4: ...reasoning hints may constrain the model's ability to explore freely, making the final performance dependent on the quality of the provided expert reasoning hints.  Q1: IGPO assumes access to accurate ground-truth reasoning traces during RL. How robust is the method to imperfect or noisy reasoning annotations? Have the authors considered evaluating IGPO with reasoning traces of varying quality to assess its sensitivity?
>
> Our response below addresses both points, supported by new robustness experiments on reasoning trace quality:
> 1. **IGPO does not restrict exploration except when exploration has already collapsed**. For prompts where the policy produces a mix of correct and incorrect samples, IGPO reduces to standard GRPO and the model explores freely. IGPO activates only in the all-wrong regime, where unrestricted on-policy exploration yields zero advantages and no learning signal. In these cases, we only injects short reasoning chunks while still allowing the policy to generate the majority (40–80%) of tokens. As shown in Fig. 5(b), IGPO improves pass@k over GRPO, indicating genuine capability gains rather than imitation of hints. IGPO's correctness gating and entropy-based filtering further prevent over-reliance on hints.
> 2. **We conducted a new robustness experiments and show IGPO is robust to noisy or imperfect reasoning traces.** To directly test sensitivity, we conduct new experiments where ground-truth traces are corrupted with realistic noise—numerical mistakes, operator swaps, logical errors, and filler tokens—whenever IGPO is triggered. Results in Appendix J (Fig. 9) show that IGPO remains effective across a wide range of noise levels: performance degrades gradually but IGPO consistently outperforms GRPO, even with 40–50% corruption. This robustness stems from two design elements: **(i) only completions that produce a correct final answer are used for learning, and (ii) only the top 20% most uncertain injected positions contribute to the gradient,** limiting the effect of erroneous hints.
>
> |  | GRPO | IGPO w/o noise | IGPO w/ 10% noise | IGPO w/ 20% noise | IGPO w/ 30% noise | IGPO w/ 40% noise | IGPO w/ 50% noise |
> | :---- | :---- | :---- | :---- | :---- | :---- | :---- | :---- |
> | Gsm8k greedy pass@1 evaluation  | 0.780 | 0.796  | 0.778 | 0.782 | 0.799 | 0.785 | 0.791 |
>
> Please refer to revision pdf’s Appendix J for detailed reward curves.

---

> ### Author Response · Authors · 2025-11-22
> **rebuttal reply (3/n)**
>
> > Question 2: Discussion or comparison between IGPO and more recent approaches addressing the zero-advantage problem, such as \[1\]?
>
> \[1\] Le, Thanh-Long V., et al. "No prompt left behind: Exploiting zero-variance prompts in llm reinforcement learning via entropy-guided advantage shaping." arXiv preprint arXiv:2509.21880 (2025).
>
> We thank the reviewer for pointing out recent work which also addresses zero-advantage prompts. Their method focuses on advantage shaping, extracting learning signal from zero-variance groups by modulating advantages with token-level entropy. We view IGPO as complementary approach: instead of reshaping advantages, we reduce zero-advantage cases **at the sampling level** by generating successful trajectories through inpainting.
>
> The two methods intervene at different stages—IGPO in exploration, RL-ZVP in credit assignment—**they could actually be combined**. And since this paper is released after the ICLR submission date, we will incorporate this related work and clarify that IGPO offers a sampling-side solution that is orthogonal to advantage-level approaches like RL-ZVP.
>
> We hope these clarifications, theoretical analysis and additional experiments address your concerns. We are happy to answer any further questions you might have.

---

> ### Author Response · Authors · 2025-11-26
> **gentle reminder**
>
> Dear reviewer raEu,
>
> Thank you again for your constructive review. In response to your suggestions, we have provided new theoretical analysis (Appendix K) and robustness experiments on noisy reasoning traces (Appendix J), along with clarified explanations of off-policyness and IGPO's correctness&entropy gating behavior. Please let us know if we can address any further concerns before the discussion period ends. Thank you!

---

### Official Review · Reviewer_NuRc · 2025-11-04

**Soundness:** 3
**Presentation:** 3
**Contribution:** 3
**Rating:** 8
**Confidence:** 3

**Summary:**

This paper proposes a policy optimization for diffusion-based LLMs that bridges supervised fine-tuning and reinforcement learning. Specifically, it introduces Inpainting Guided Policy Optimization (IGPO), which leverages the unique inpainting capabilities of full-attention dLLMs by strategically injecting partial ground-truth reasoning traces into online policy optimization. With this design as well as some other techniques such as entropy-based filtering, they achieve better sample efficiency than GRPO and achieve SOTA among full-attention masked dLLMs across four mathematical reasoning benchmarks.

**Strengths:**

+ The paper presents the first work to utilize the unique inpainting capabilities of diffusion LLMs
for reinforcement learning, which is novel.
+ The method achieves SOTA results on four benchmarks for full-attention based dLLMs.
+ The ablation studies demonstrate insights of the proposed method, such as how self-generated inpainted traces provide a better learning signal than ground truth traces.

**Weaknesses:**

The proposed method is a hybrid between supervised learning (or imitation learning in the general RL field) and online reinforcement learning. The authors should compare or at least discuss other guided exploration methods in the general RL domain (especially in areas like diffusion policy) to give more insights into how this method is particularly good for dLLMs and how it might be or might not be transferable to tuning other diffusion-based models (say, video diffusion models or diffusion policies). Admittedly, the full-attention dLLMs have their unique properties.

**Questions:**

See weaknesses

---

> ### Author Response · Authors · 2025-11-22
>
> Dear reviewer NuRc ,
>
> We are thankful for your time and for the helpful suggestion. We were glad to hear that you found our application of inpainting to RL novel and our results SOTA among full-attention dLLMs. Below we address your concern:
>
> > Weakness 1:  The authors should compare or at least discuss other guided exploration methods in the general RL domain (especially in areas like diffusion policy) to give more insights into how this method is particularly good for dLLMs and how it might be or might not be transferable to tuning other diffusion-based models (say, video diffusion models or diffusion policies).
>
> We agree that placing IGPO in the broader context of guided exploration is valuable. We will add a discussion in related work comparing guided exploration methods in diffusion-based RL policies.
>
> One notable diffusion-based RL policy work is Diffuser[1]. Diffuser uses classifier/critic guidance to modify the sampling distribution at test time, steering the reverse diffusion process toward high-reward or goal-reaching trajectories. Importantly, this guidance is not used during training: the diffusion model is trained purely via supervised behavioral cloning on offline trajectories, and guidance only perturbs the denoising process at inference.
>
> This differs from IGPO in that IGPO modifies the online policy optimization process itself, not the test-time sampler. Rather than injecting continuous reward gradients or critic scores, IGPO introduces semantic, discrete partial reasoning traces during RL sampling — the only form of “guidance’’ that is meaningful for language. Moreover, IGPO activates guidance only in zero-advantage groups, addressing the exploration-collapse failure mode in GRPO, whereas Diffuser’s guidance is unconditional and only applies at inference time.
>
> >  how it might be or might not be transferable to tuning other diffusion-based models
>
> While IGPO is specifically designed for discrete, textual reasoning in dLLMs, the underlying idea—using partial hint injection to provide targeted guidance when exploration collapses—could in principle, transfer to other diffusion policies like Diffuser or multimodal dLLM such as MMaDA[2]. For continuous trajectory diffusion models, the analogue of “partial reasoning traces’’ would be partial trajectory constraints (e.g., fixing key waypoints or sub-goal segments) that are injected only when rollout failures occur.
>
>  Exploring such an extension is an interesting future direction, though domain-specific adaptations would be required.
>
> [1] Planning with Diffusion for Flexible Behavior Synthesis. arXiv:2205.09991
> [2] MMaDA: Multimodal large diffusion language models. arXiv:2505.15809.

---

### Author Response · Authors · 2025-11-26
**Rebuttal summary for reviewers and ACs**

We sincerely thank all reviewers for their thoughtful feedback and constructive suggestions, which have helped us improve our paper through clarifications, theoretical analysis, and new experiments in the revised paper. Below we summarize the main concerns and our responses:

**1. Theoretical justification (Reviewers L2eU, RaEu)**

We added in Appendix K in the revised paper proving two theorems to explain why inpainting improves RL optimization in all-wrong cases.

* Theorem 1: Gradient Recovery & Policy Improvement:
  In the all-wrong, zero-advantage regime where GRPO provably yields zero gradient, IGPO restores a non-degenerate gradient by replacing a small fraction of responses with correctness-verified inpainted completions. And under a mixture-policy view, IGPO strictly improves expected reward whenever the base policy is imperfect.

* Theorem 2: KL Control of Partial Inpainting:
  We prove that injecting only a fraction of ground truth tokens induces a KL shift that scales linearly with the inpainting ratio, giving a soft trust-region–like bound. This explains IGPO’s stability and why partial hints outperform full-hint or SFT-based alternatives in our experiments.



**2. Off-Policy concern (Reviewers LgBR, RaEu)**

We clarified that although hint tokens are technically off-policy, IGPO maintains near-on-policy behavior for the following reasons:

* Most tokens (~80%) remain model-generated.
Hint tokens form a small fraction of each trajectory, and only appear when exploration has already collapsed.

* Two gates suppress instability: (1) Entropy-based gradient filtering (only the top 20% most uncertain hint positions contribute gradients). (2) Importance-ratio clipping, for which we empirically observe the actual clipping rate to be near zero across training, indicating that hint tokens lie inside the clipped region and do not destabilize learning.
* Our added theoretical analysis shows that the distribution shift induced by partial inpainting is KL-controlled and strictly beneficial in the zero-advantage case, where on-policy GRPO provides no learning signal.


**3. Robustness to noisy reasoning traces (Reviewers RaEu, L2eU)**

We conducted new experiments (in Appendix J) where reasoning traces used for inpainting are corrupted with numeric, operator, logical, and text-noise perturbations up to 50% token-level corruption. Result shows that IGPO remains robust and continues to outperform GRPO even under heavy noise. The stability arises from IGPO’s dual gating mechanisms as mentioned in point 2 above: correctness filtering and entropy-based token selection. IGPO also only requires reasoning traces on questions the current LLM can not solve.


**4. Compare IGPO with sample size-matched GRPO (Reviewer LgBR)**

We added a new sample-matched baseline: “GRPO + resample” that draws the same number of additional samples as IGPO, without hint injection. As shown in reward curves, all-wrong ratio, and final accuracy (Appendix L)—IGPO consistently outperforms resampling.
This shows the gains come from hint injection, not from increased sampling compute.

**5. Comparison to distillation / SFT-based baselines (Reviewer LgBR)**

We compared IGPO against: (1) SFT on the same gold reasoning traces for inpainting, and (2) SFT followed by GRPO. We show that both underperform IGPO (Appendix E & Fig. 7). Notably, concise MetaMath-style traces used for inpainting hurt performance when used for SFT, but help when used through IGPO’s partial, correctness-gated injection. dLLM's inpainting ability can help convert low quality traces to correct traces through injecting partial correct tokens.

---

### Meta-Review · Area_Chair_ZgFt · 2026-01-07

**Summary:**

Reviewers found the problem setting timely and the empirical results strong, but raised concerns regarding (i) whether IGPO fundamentally differs from entropy regularization or supervised fine-tuning on reasoning traces, (ii) the validity of introducing partial ground-truth tokens in an ostensibly on-policy RL framework, (iii) the dependence on high-quality reasoning traces and diffusion-specific architectures, and (iv) the clarity and rigor of the theoretical justification. Some reviewers also asked whether observed gains could be attributed to increased sampling or compute rather than the proposed inpainting mechanism, and how IGPO relates to other guided exploration or zero-advantage mitigation methods.

**Reviewer Concerns:**

Addressed concerns

- Attribution of gains beyond increased sampling: The authors added compute-matched baselines (e.g., GRPO with resampling) and showed that IGPO outperforms these alternatives, supporting that improvements are not simply due to additional sampling.

- Comparison to supervised fine-tuning and distillation: Additional experiments comparing IGPO to SFT on the same reasoning traces and SFT followed by GRPO showe that IGPO provides superior performance.

- Clarity and theoretical justification: The method description was revised, and new theoretical results were added (Appendix K) analyzing gradient recovery in zero-advantage regimes and relating the inpainting ratio to KL-based trust-region behavior.

- Robustness to imperfect reasoning traces: New experiments with noisy and partially corrupted traces show that IGPO remains effective even when injected reasoning is imperfect.

- On-policy validity in practice: The authors clarified that while IGPO departs from a strictly textbook on-policy formulation, this is consistent with how PPO-style methods are used in practice. Most tokens remain policy-generated, gradients are clipped and KL-regularized, and empirical diagnostics (e.g., low clipping rates) indicate controlled distributional shift. This reframes the issue as one of guided exploration rather than invalid optimization.

Outstanding concerns

- Scope and generality: IGPO is specific to diffusion-based LLMs and requires access to ground-truth or high-quality reasoning traces, which limits applicability to certain domains and datasets.

- Positioning relative to concurrent guided-exploration methods: Comparisons to some contemporaneous approaches are discussed rather than experimentally evaluated, leaving room for future empirical clarification.

**Reviewer Scores:**

- Reviewer NuRc: No change (8 → 8). Already positive; rebuttal clarifications are consistent with the reviewer’s assessment.
- Reviewer LgBR: Upward change (4 → 6). The reviewer explicitly stated that the additional experiments and analyses addressed their concerns and that they advocate acceptance.
- Reviewer L2eU: Possible upward change (4 → 6). The main blockers were clarity and theory, which were directly addressed in the rebuttal; a positive score revision is plausible.
- Reviewer raEu: No change (4 → 4). Core concerns about generality and reliance on reasoning traces remain valid and are not fully resolved. I assume no change in score.

---

### Decision · Program_Chairs · 2026-01-26

Accept (Poster)